# TEXT4SEG: REIMAGINING IMAGE SEGMENTATION AS TEXT GENERATION

**Mengcheng Lan, Chaofeng Chen, Yue Zhou**
S-Lab, Nanyang Technological University
lanm0002@e.ntu.edu.sg
{chaofeng.chen,yue.zhou}@ntu.edu.sg

**Jiaxing Xu, Yiping Ke**
CCDS, Nanyang Technological University
jiaxing003@e.ntu.edu.sg
ypke@ntu.edu.sg

**Xinjiang Wang, Litong Feng**[*]**,Wayne Zhang**
SenseTime Research
{wangxinjiang,fenglitong,wayne.zhang}@sensetime.com

## ABSTRACT

Multimodal Large Language Models (MLLMs) have shown exceptional capabilities in vision-language tasks; however, effectively integrating image segmentation into these models remains a significant challenge. In this paper, we introduce Text4Seg, a novel *text-as-mask* paradigm that casts image segmentation as a text generation problem, eliminating the need for additional decoders and significantly simplifying the segmentation process. Our key innovation is *semantic descriptors*, a new textual representation of segmentation masks where each image patch is mapped to its corresponding text label. This unified representation allows seamless integration into the auto-regressive training pipeline of MLLMs for easier optimization. We demonstrate that representing an image with $16 \times 16$ semantic descriptors yields competitive segmentation performance. To enhance efficiency, we introduce the Row-wise Run-Length Encoding (R-RLE), which compresses redundant text sequences, reducing the length of semantic descriptors by 74% and accelerating inference by $3\times$, without compromising performance. Extensive experiments across various vision tasks, such as referring expression segmentation and comprehension, show that Text4Seg achieves state-of-the-art performance on multiple datasets by fine-tuning different MLLM backbones. Our approach provides an efficient, scalable solution for vision-centric tasks within the MLLM framework. https://github.com/mc-lan/Text4Seg

## 1 INTRODUCTION

Multimodal Large Language Models (MLLMs) (Yin et al., 2023) have successfully extended the capabilities of powerful Large Language Models (LLMs) into the visual domain. Recent advancements demonstrate the remarkable ability of these models to engage in natural language-based human-computer interaction and text-based reasoning over visual inputs (Liu et al., 2024c; Lu et al., 2024; Liu et al., 2024a; Bai et al., 2023; Chen et al., 2024). MLLMs have emerged as powerful tools for vision-centric tasks, including image generation (Song et al., 2024; Wang et al., 2024d), object detection (Wang et al., 2024a; Ma et al., 2024; Zhang et al., 2023) and semantic segmentation (Lai et al., 2024; Zhang et al., 2024b; Lan et al., 2024c). However, seamlessly integrating MLLMs with these tasks, particularly in dense prediction tasks like semantic segmentation, remains challenging due to the intrinsic differences between language and visual modalities.

A straightforward approach adopted by most existing works (Lai et al., 2024; Xia et al., 2024; Zhang et al., 2024b; He et al., 2024; Ren et al., 2024; Rasheed et al., 2024; Wang et al., 2024c; Zhang et al., 2023; Wu et al., 2024) involves appending additional visual decoders (*e.g.*, SAM (Kirillov et al., 2023)) to MLLMs, as illustrated in Fig. 1(a). While effective, this combination presents several limitations: 1) it complicates the end-to-end training pipeline with additional loss functions; 2) it requires careful modifications to MLLM architectures, leading to unexpected challenges when scaling

---
[*]Corresponding author.

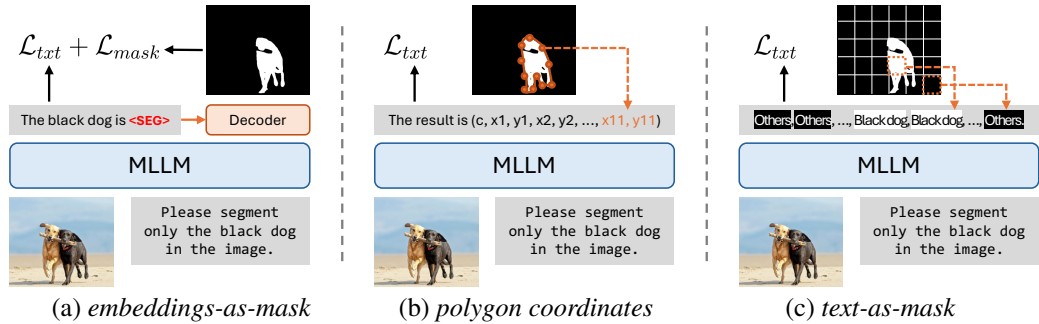

(a) *embeddings-as-mask*    (b) *polygon coordinates*    (c) *text-as-mask*

Figure 1: Different paradigms of MLLMs based image segmentation: (a) *embeddings-as-mask* paradigm that relies on additional segmentation decoder and loss (*e.g.*, LISA (Lai et al., 2024)); (b) *polygon coordinates* for instance segmentation (*e.g.*, VisionLLM (Wang et al., 2024a)); (c) our *text-as-mask* paradigm that relies on semantically consistent text sequences.

up the training. VisionLLM (Wang et al., 2024a) attempts to convert segmentation masks into polygon coordinate sequences, as shown in Fig. 1(b). However, the performance is often unsatisfactory, as LLMs may struggle to associate polygon coordinates with shapes, leading to the reintroduction of segmentation-specific decoders in VisionLLMv2 (Jiannan et al., 2024). Finding a more effective method to unlock the segmentation capabilities for MLLMs remains crucial. Such method should adhere to the next-token prediction paradigm of MLLMs for easier optimization, require fewer architectural changes for better scalability, and fully leverage text generation capabilities of LLMs.

In this paper, we introduce a novel *text-as-mask* paradigm that casts image segmentation as a text generation problem, which significantly simplifies the segmentation process. We propose **Text4Seg**, a decoder-free framework for MLLMs based image segmentation, as illustrated in Fig. 1(c). Central to our method is a novel sequence representation of segmentation masks. Instead of using index masks or numerical coordinates, we map each flattened patch of the input image to its corresponding text description (*e.g.*, a semantic label, a short phrase, or a long sentence), forming a purely textual representation of images, named as **semantic descriptors**. This representation offers several advantages: 1) a unified sequence representation seamlessly integrated into the auto-regressive training pipeline, making joint optimization with text tasks easier; 2) no architectural changes are required, allowing full utilization of existing MLLM training infrastructure, making it ideal for scaling up; 3) support for large label vocabularies, equivalent to semantic words; and 4) flexible switching between referring expression segmentation, open-vocabulary segmentation, and other visual grounding tasks.

Inspired by ViT (Dosovitskiy et al., 2021), we demonstrate that *representing an image with $16 \times 16$ semantic words, i.e.,* 256 *length of semantic descriptors, is sufficient to achieve satisfactory results*. To improve efficiency, we introduce the Row-wise Run-Length Encoding (R-RLE), which compresses the repeated descriptors within each image row while preserving the spatial structure. *Without compromising performance*, R-RLE achieves a 74% reduction in semantic descriptors length and speeds up inference by $3\times$ on average. To further enhance performance, we apply an off-the-shelf mask refiner, *i.e.*, SAM, as a post-processing method to obtain pixel-level segmentation masks.

With the proposed semantic descriptors, training MLLMs for segmentation requires minimal additional effort. We begin by constructing instruction-following data from existing segmentation datasets, transforming the vanilla semantic masks into the semantic descriptors format, and then fine-tuning the model using query-response conversations. This approach applies to a variety of vision-centric tasks, such as referring expression segmentation, open-vocabulary segmentation, and visual grounding tasks. Our experiments demonstrate that Text4Seg can seamlessly integrate segmentation capabilities into existing MLLM architectures, such as LLaVA-1.5 (Li et al., 2024a), Qwen-VL (Bai et al., 2023), DeepseekVL (Lu et al., 2024), and InternVL2 (Chen et al., 2023b), *without any architectural modifications*. Without bells and whistles, Text4Seg consistently achieves superior or comparable performance to previous models, highlighting its efficiency, flexibility, and robustness. In summary, our key contributions are as follows:

- We propose Text4Seg, a novel *text-as-mask* paradigm that redefines image segmentation as a text generation problem, fully leveraging the text generation capabilities of MLLMs.

- We introduce semantic descriptors, a textual sequence representation of segmentation masks that seamlessly integrates with existing MLLMs for easier optimization. We demonstrate that $16 \times 16$ semantic descriptors are sufficient for achieving strong performance.

- We develop Row-wise Run-Length Encoding (R-RLE) to compress semantic descriptors, significantly reducing its length and inference costs without compromising performance.

- We validate the effectiveness and robustness of Text4Seg based on various MLLMs backbones by achieving state-of-the-art performance across various vision centric tasks.

## 2 RELATED WORK

**Multimodal Large Language Models.** MLLMs are typically developed by enhancing large language models (LLMs) with visual perception modules, which can generate coherent textual conversations grounded in multimodal inputs. For instance, Flamingo (Alayrac et al., 2022) introduces the Perceiver Resampler, which connects a pre-trained vision encoder with LLMs for effective few-shot learning. OpenFlamingo (Awadalla et al., 2023) and Otter (Li et al., 2023a) build upon this architecture with a focus on multi-modal in-context instruction tuning. BLIP-2 (Li et al., 2023b) and InstructBLIP (Dai et al., 2023) bridge the modality gap using a lightweight Querying Transformer (Q-Former), demonstrating enhanced performance on zero-shot vision-to-language tasks. The LLaVA seires (Liu et al., 2024c;a) employs a linear layer or MLP as a modality connector, trained on multimodal language-image instruction-following data generated with GPT-4, showcasing notable capabilities in multimodal chat interactions. They demonstrate impressive capabilities in multimodal chat interactions. Recent advancements (Liu et al., 2024b; Xu et al., 2024; Li et al., 2024a;b;c; Lin et al., 2023) have focused on enhancing visual encoding through high-resolution inputs. For example, LLaVA-UHD (Xu et al., 2024) implements an image modularization strategy, segmenting native-resolution images into smaller, variable-sized slices to improve scalability and encoding efficiency. Similarly, LLaVA-NEXT (Liu et al., 2024b) and LLaVA-OneVision (Li et al., 2024a) utilize the AnyRes scheme to accommodate high-resolution image inputs. In this work, we present Text4Seg to endow existing MLLMs with image segmentation capabilities based on instruction tuning, *without necessitating any changes to their architecture*.

**Language-Guided Semantic Segmentation and Localization.** Recent advancements have enabled MLLMs to incorporate task-specific modules for vision-centric tasks. LISA (Lai et al., 2024) introduces the embedding-as-mask paradigm, utilizing a special <seg> token to prompt a segmentation mask decoder, such as SAM (Kirillov et al., 2023), thereby enhancing performance in reasoning and referring expression segmentation. Building on this, GSVA (Xia et al., 2024) employs multiple <seg> tokens and a <REJ> token to address cases where users reference multiple subjects or provide descriptions mismatched with image targets. Similarly, GLaMM (Rasheed et al., 2024) extends LISA's single-object focus by integrating natural language responses with corresponding object segmentation masks. They introduce a large-scale, densely annotated Grounding-anything Dataset to train GLaMM, which significantly improves performance across various vision tasks. OMG-LLaVA (Zhang et al., 2024a) and PixelLM (Ren et al., 2024) are also capable of grounded conversation generation. PixelLM (Ren et al., 2024) advances LISA further by replacing SAM with a lightweight pixel decoder and introducing a comprehensive segmentation codebook for efficient multi-target reasoning and segmentation. In contrast, GROUNDHOG (Zhang et al., 2024b) proposes inputting visual entity tokens, rather than visual tokens, using their masked feature extractor, which enables fine-grained visual understanding. GROUNDHOG also curated a grounded visual instruction tuning dataset with Multi-Modal Multi-Grained Grounding, M3G2, to fully train the model. Recent studies (Zhang et al., 2023; Jiannan et al., 2024; Wu et al., 2024; Fei et al., 2024) extend MLLMs to vision-centric tasks like visual grounding (*e.g.*, bounding boxes, masks) by integrating task-specific heads for different applications. While effective, these approaches increase training complexity and limit model scalability due to multiple decoders and loss functions. Other efforts (Chen et al., 2021; Peng et al., 2023; Wang et al., 2024a) have sought to simplify this process by learning coordinate sequences or location tokens. However, they tend to perform well only in object detection tasks with simple location coordinates, and struggle to achieve competitive results on more complex tasks such as segmentation. In contrast, we introduce a general sequence representation for vision tasks without task-specific heads, enabling seamless integration with MLLMs and leveraging their text-generation capabilities for effective, versatile performance across applications.

## 3 METHODOLOGY

### 3.1 PRELIMINARY

Multimodal Large Language Models (MLLMs) (Yin et al., 2023) refer to the LLM-based models with the ability to process, reason, and generate response from multimodal information. Typically, as shown in Fig. 2, an MLLM can be abstracted into three main components: 1) a pre-trained vision encoder, which is responsible for extracting visual tokens from input images, 2) a pre-trained large language model (LLM), which handles reasoning and generating outputs, and 3) a modality connector, which acts as a bridge between the vision encoder and the LLM.

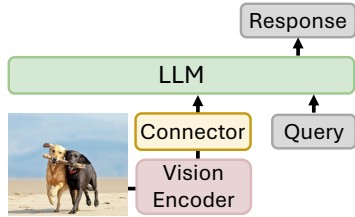

Figure 2: MLLM architecture.

### 3.2 SEMANTIC DESCRIPTORS

**Definition of semantic descriptors.** Our semantic descriptors are inspired by ViT (Dosovitskiy et al., 2021), which represents an image as $16 \times 16$ visual tokens. As illustrated in Fig. 3, for simplicity, the example uses $6 \times 6$ visual tokens, the process begins by splitting the image into fixed-size patches and flattening them. Each patch is then represented by its corresponding semantic descriptor. A descriptor can be as simple as a semantic label (*e.g.*, "sky," "sand"), a phrase (*e.g.*, "brown dog", "black dog"), or even a more complex textual description (*e.g.*, "a dog in the left") for intricate scenes. This approach encodes an image into a sequence of semantic descriptors of length 256, which meets the requirements for integrating image segmentation into MLLMs by:

- Adhering to the next-token prediction paradigm of MLLMs, facilitating easier optimization.

- Requiring no architectural changes, ensuring seamless integration and scalability.

- Adopting a text-as-mask paradigm, using text generation capabilities of LLMs for segmentation.

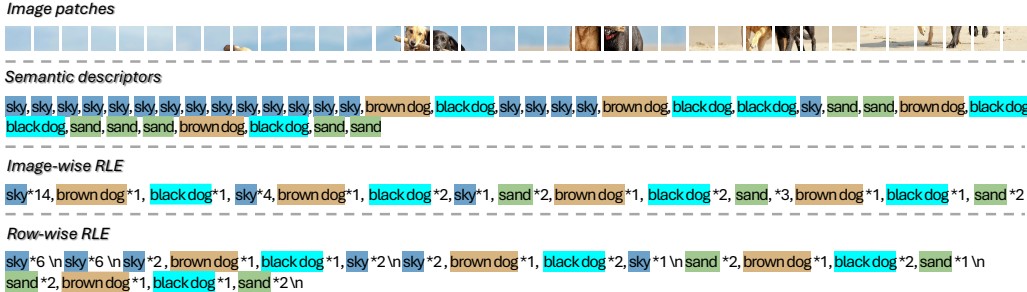

Figure 3: An illustration of semantic descriptors for images and two token compression techniques.

**Row-wise RLE.** One of the key limitations of full-length semantic descriptors is the long token length due to the inherent spatial redundancy in images. For instance, the average token length of 256 semantic descriptors on the refCOCO (Kazemzadeh et al., 2014) dataset is 583, requiring approximately 19s on a V100 GPU for a single round of referring expression segmentation. To address this issue, we introduce the simple Run-Length Encoding (RLE) (Golomb, 1966) to compress the adjacent repeated texts in semantic descriptors.

A straight forward approach is to directly apply RLE to the whole semantic descriptors, referred as Image-wise RLE (I-RLE). However, we empirically found that it results in a notable performance drop, suggesting that the compressed descriptors may lose crucial spatial information.

To mitigate this issue, we propose a novel Row-wise Run-Length Encoding (R-RLE) technique. As shown in Fig. 3, R-RLE operates at the row level, with each row separated by "\n". This approach reduces the token length from 583 to 154 on average while preserving more spatial information.

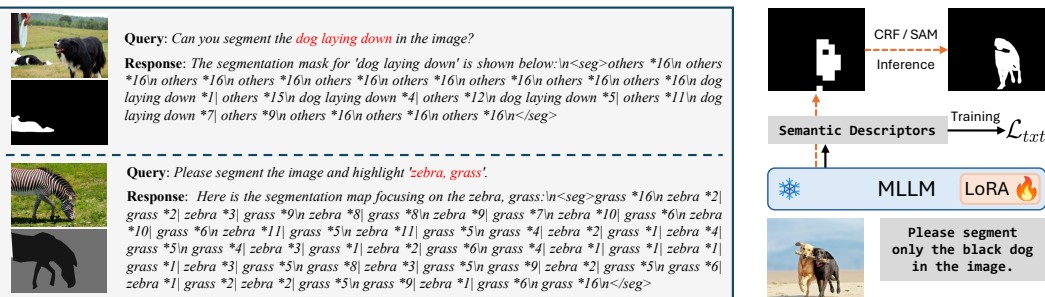

Figure 4: Visual instruction data.          Figure 5: Text4Seg.

Importantly, R-RLE demonstrates no performance degradation compared to the full-length semantic descriptors, and significantly enhances the inference speed.

### 3.3 VISUAL INSTRUCTION TUNING OF TEXT4SEG

Building on the proposed semantic descriptors, we construct visual instruction data by leveraging existing segmentation datasets. Fig. 4 shows examples for referring expression segmentation and semantic segmentation. Given a pair of <image, mask>, we resize the mask to a $16 \times 16$ resolution and flatten it. The indexes in the sequence are then replaced with their corresponding text labels to create full-length semantic descriptors. We further apply R-RLE to compress the sequence, with descriptors separated by "|" and rows separated by "\n". Finally, the image, text labels, and semantic descriptors are embedded into a query-response template like

**Query:** <IMAGE> Can you segment the <text labels> in the image?
**Response:** The result is :\n <seg>semantic descriptors< /seg>.

Note that <seg> and < /seg> are start and end of semantic descriptors. With such pure text response, Text4Seg can be seamlessly integrated with existing MLLMs without any architectural modifications, as shown in Fig. 5. We use Low-Rank Adaptation (LoRA) (Hu et al., 2021), to fine-tune the MLLMs on our visual instruction data, using its original auto-regressive training objective $\mathcal{L}_{txt}$. In contrast to existing models (Lai et al., 2024; Zhang et al., 2024b; Rasheed et al., 2024), which typically rely on Continued Pre-Training (CPT) with large, mixed datasets to fuse the architectures before fine-tuning on specific downstream tasks, we apply Supervised Fine-Tuning (SFT) directly on the downstream tasks. During inference, the coarse masks generated by the MLLM demonstrate competitive performance compared to existing methods. To enhance the quality of pixel-level semantic masks, we optionally apply either Conditional Random Fields (CRF) (Krähenbühl & Koltun, 2011) or the SAM as the mask refiner.

## 4 EXPERIMENTS

### 4.1 IMPLEMENTATION DETAILS

**Model architectures.** Our method is built upon several open-source MLLMs, including LLaVA-1.5 (Liu et al., 2024a), DeepseekVL (Lu et al., 2024), InternVL2 (Chen et al., 2024), and Qwen-VL (Bai et al., 2023). The main experiments cover 6 MLMMs with model sizes ranging from 1.3B to 13B parameters, and 3 connectors, including MLP (LLaVA-1.5, DeepseekVL), Pixel Shuffle + MLP (InternVL2) and Cross-attention (Qwen-VL). All architectures were left unaltered during the experiments. Additionally, we employ the off-the-shelf SAM with ViT-H as our mask refiner.

**Model training.** Our method is implemented using SWIFT (Zhao et al., 2024). All models are trained on 8 Tesla A800 GPUs (40GB) with a global batch size of 128. We use the AdamW optimizer (Loshchilov, 2017), starting with an initial learning rate of 2e-4, which follows a linear decay schedule after a warm-up phase with a ratio of 0.03. The weight decay is set to 0, and gradient norms are clipped at 1.0. To minimize GPU memory usage, we fine-tune all models using LoRA with a rank of 64, along with ZeRO-2 stage memory optimization.

Table 1: **Referring Expression Segmentation** results (cIoU) on refCOCO (+/g) datasets (Kazemzadeh et al., 2014; Mao et al., 2016). GLaMM is depicted in a lighter color as it uses a training dataset two orders of magnitude larger than ours. † Model with CRF as the mask refiner. ‡ Model based on the 32× 32 semantic descriptors without the mask refiner.

| Methods | LLM | refCOCO | | | refCOCO+ | | | refCOCOg | | Avg. |
|---|---|---|---|---|---|---|---|---|---|---|
| | | val | testA | testB | val | testA | testB | val | test | |
| *Specialised Segmentation Models* | | | | | | | | | | |
| ReLA (Liu et al., 2023a) | | 73.8 | 76.5 | 70.2 | 66.0 | 71.0 | 57.7 | 65.0 | 66.0 | 68.3 |
| HIPIE (Wang et al., 2024b) | | 78.3 | - | - | 66.2 | - | - | 69.8 | - | - |
| PolyFormer-L (Liu et al., 2023b) | | 76.0 | 78.3 | 73.3 | 69.3 | 74.6 | 61.9 | 69.2 | 70.2 | 71.6 |
| UNINEXT-L (Yan et al., 2023) | | 80.3 | 82.6 | 77.8 | 70.0 | 74.9 | 62.6 | 73.4 | 73.7 | 74.4 |
| *Generalist Segmentation Models (≤8B)* | | | | | | | | | | |
| NEXT-Chat (Zhang et al., 2023) | Vicuna-7B | 74.7 | 78.9 | 69.5 | 65.1 | 71.9 | 56.7 | 67.0 | 67.0 | 68.9 |
| LISA (Lai et al., 2024) | Vicuna-7B | 74.9 | 79.1 | 72.3 | 65.1 | 70.8 | 58.1 | 67.9 | 70.6 | 69.9 |
| PixelLM (Ren et al., 2024) | Vicuna-7B | 73.0 | 76.5 | 68.2 | 66.3 | 71.7 | 58.3 | 69.3 | 70.5 | 69.2 |
| AnyRef (He et al., 2024) | LLaMA2-7B | 76.9 | 79.9 | 74.2 | 70.3 | 73.5 | 61.8 | 70.0 | 70.7 | 72.2 |
| GSVA (Xia et al., 2024) | Vicuna-7B | 77.2 | 78.9 | 73.5 | 65.9 | 69.6 | 59.8 | 72.7 | 73.3 | 71.4 |
| LaSagnA (Wei et al., 2024) | Vicuna-7B | 76.8 | 78.7 | 73.8 | 66.4 | 70.6 | 60.1 | 70.6 | 71.9 | 71.1 |
| Groundhog (Zhang et al., 2024b) | LLaMA2-7B | 78.5 | 79.9 | 75.7 | 70.5 | 75.0 | 64.9 | 74.1 | 74.6 | 74.2 |
| GLaMM (Rasheed et al., 2024) | Vicuna-7B | 79.5 | 83.2 | 76.9 | 72.6 | 78.7 | 64.6 | 74.2 | 74.9 | 75.6 |
| Text4Seg DeepseekVL-1.3B | DeepSeek-1.3B | 75.0 | 78.6 | 70.1 | 68.4 | 73.4 | 60.0 | 71.5 | 71.7 | 71.1 |
| Text4Seg DeepseekVL-7B† | DeepSeek-7B | 72.6 | 74.8 | 70.0 | 67.2 | 71.5 | 62.2 | 69.1 | 69.4 | 69.6 |
| Text4Seg DeepseekVL-7B | DeepSeek-7B | 78.8 | 81.5 | 74.9 | 72.5 | 77.4 | 65.9 | 74.3 | 74.4 | 75.0 |
| Text4Seg Qwen-VL-7B† | Qwen-7B | 71.3 | 73.7 | 69.6 | 65.9 | 70.4 | 61.9 | 69.3 | 69.3 | 68.9 |
| Text4Seg Qwen-VL-7B | Qwen-7B | 78.0 | 80.9 | 74.6 | 71.6 | 77.3 | 66.0 | **74.8** | 74.7 | 74.7 |
| Text4Seg LLaVA-1.5-7B† | Vicuna-7B | 73.2 | 75.7 | 71.4 | 67.0 | 71.9 | 62.4 | 67.3 | 68.9 | 69.7 |
| Text4Seg LLaVA-1.5-7B | Vicuna-7B | **79.3** | **81.9** | **76.2** | 72.1 | 77.6 | 66.1 | 72.1 | 73.9 | 74.9 |
| Text4Seg InternVL2-8B† | InternLM2.5-7B | 73.0 | 75.2 | 70.7 | 67.6 | 72.1 | 62.6 | 68.9 | 70.3 | 70.1 |
| Text4Seg InternVL2-8B‡ | InternLM2.5-7B | 74.7 | 77.4 | 71.6 | 68.5 | 73.6 | 62.9 | 70.7 | 71.6 | 71.4 |
| Text4Seg InternVL2-8B | InternLM2.5-7B | 79.2 | 81.7 | 75.6 | **72.8** | **77.9** | **66.5** | 74.0 | **75.3** | **75.4** |
| *Generalist Segmentation Models (13B)* | | | | | | | | | | |
| LISA (Lai et al., 2024) | Vicuna-13B | 76.0 | 78.8 | 72.9 | 65.0 | 70.2 | 58.1 | 69.5 | 70.5 | 70.1 |
| GSVA (Xia et al., 2024) | Vicuna-13B | 78.2 | 80.4 | 74.2 | 67.4 | 71.5 | 60.9 | **74.2** | **75.6** | 72.8 |
| Text4Seg LLaVA-1.5-13B† | Vicuna-13B | 74.1 | 76.4 | 72.4 | 68.5 | 72.8 | 63.6 | 69.1 | 70.1 | 70.9 |
| Text4Seg LLaVA-1.5-13B | Vicuna-13B | **80.2** | **82.7** | **77.3** | **73.7** | **78.6** | **67.6** | 74.0 | 75.1 | **76.2** |

Table 2: **Generalized Referring Expression Segmentation** results on the grefCOCO dataset (Liu et al., 2023a). † Model with CRF as the mask refiner. ‡ Model based on the 32× 32 semantic descriptors without the mask refiner.

| Methods | LLM | Validation Set | | Test Set A | | Test Set B | | Avg. |
|---|---|---|---|---|---|---|---|---|
| | | gIoU | cIoU | gIoU | cIoU | gIoU | cIoU | |
| *Specialised Segmentation Models* | | | | | | | | |
| LAVT (Yang et al., 2022) | | 58.4 | 57.6 | 65.9 | 65.3 | 55.8 | 55.0 | 59.7 |
| ReLA (Liu et al., 2023a) | | 63.6 | 62.4 | 70.0 | 69.3 | 61.0 | 59.9 | 64.4 |
| *Generalist Segmentation Models (≤8B)* | | | | | | | | |
| LISA (Lai et al., 2024) | Vicuna-7B | 61.6 | 61.8 | 66.3 | 68.5 | 58.8 | 60.6 | 62.9 |
| GSVA (Xia et al., 2024) | Vicuna-7B | 66.5 | 63.3 | 71.1 | 69.9 | 62.2 | 60.5 | 65.6 |
| Text4Seg DeepseekVL-1.3B | DeepSeek-1.3B | 69.9 | 63.2 | 69.7 | 67.5 | 62.3 | 59.8 | 65.4 |
| Text4Seg DeepseekVL-7B† | DeepSeek-7B | 70.4 | 65.8 | 68.9 | 69.9 | 63.2 | 63.6 | 67.0 |
| Text4Seg DeepseekVL-7B | DeepSeek-7B | **74.7** | 69.0 | 74.3 | 73.0 | 67.4 | 66.3 | 70.8 |
| Text4Seg Qwen-VL-7B† | Qwen-7B | 69.7 | 64.1 | 67.4 | 67.8 | 62.4 | 62.3 | 65.6 |
| Text4Seg Qwen-VL-7B | Qwen-7B | 74.4 | 68.1 | 73.1 | 71.5 | 66.7 | 65.3 | 69.9 |
| Text4Seg LLaVA-1.5-7B† | Vicuna-7B | 69.1 | 64.7 | 69.9 | 70.8 | 62.1 | 62.3 | 66.5 |
| Text4Seg LLaVA-1.5-7B | Vicuna-7B | 73.6 | 67.9 | 74.1 | 72.8 | 66.1 | 64.8 | 69.9 |
| Text4Seg InternVL2-8B† | InternLM2.5-7B | 70.0 | 66.1 | 69.4 | 70.9 | 63.1 | 64.1 | 67.3 |
| Text4Seg InternVL2-8B‡ | InternLM2.5-7B | 71.8 | 65.6 | 71.2 | 70.0 | 64.2 | 62.5 | 67.6 |
| Text4Seg InternVL2-8B | InternLM2.5-7B | 74.4 | **69.1** | **75.1** | 73.8 | **67.3** | **66.6** | **71.1** |
| *Generalist Segmentation Models (13B)* | | | | | | | | |
| LISA (Lai et al., 2024) | Vicuna-13B | 63.5 | 63.0 | 68.2 | 69.7 | 61.8 | 62.2 | 64.7 |
| GSVA (Xia et al., 2024) | Vicuna-13B | 68.0 | 64.1 | 71.8 | 70.5 | 63.8 | 61.3 | 66.6 |
| Text4Seg LLaVA-1.5-13B† | Vicuna-13B | 70.3 | 66.9 | 69.8 | 71.4 | 63.8 | 64.4 | 67.8 |
| Text4Seg LLaVA-1.5-13B | Vicuna-13B | **74.8** | **69.8** | **75.1** | **74.3** | **68.0** | **67.1** | **71.5** |

## 4.2 REFERRING EXPRESSION SEGMENTATION

**Settings.** For referring expression segmentation (RES), we follow standard evaluation protocols (Lai et al., 2024; Xia et al., 2024) and assess our method using the refCOCO series. We construct

Table 3: **Referring Expression Comprehension** results (Acc@0.5) on RefCOCO (+/g) datasets (Kazemzadeh et al., 2014; Mao et al., 2016). * Model without the mask refiner.

| Methods | LLM | refCOCO | | | refCOCO+ | | | refCOCOg | | Avg. |
|---|---|---|---|---|---|---|---|---|---|---|
| | | val | testA | testB | val | testA | testB | val | test | |
| *Specialised Segmentation Models* | | | | | | | | | | |
| MDETR (Kamath et al., 2021) | | 86.8 | 89.6 | 81.4 | 79.5 | 84.1 | 70.6 | 81.6 | 80.9 | 81.8 |
| G-DINO (Liu et al., 2023c) | | 90.6 | 93.2 | 88.2 | 82.8 | 89.0 | 75.9 | 86.1 | 87.0 | 86.6 |
| PolyFormer-L (Liu et al., 2023b) | | 90.4 | 92.9 | 87.2 | 85.0 | 89.8 | 78.0 | 85.8 | 85.9 | 86.9 |
| UNINEXT-L (Yan et al., 2023) | | 91.4 | 93.7 | 88.9 | 83.1 | 87.9 | 76.2 | 86.9 | 87.5 | 87.0 |
| *Generalist Segmentation Models ($\leq$8B)* | | | | | | | | | | |
| Shikra (Chen et al., 2023a) | Vicuna-7B | 87.0 | 90.6 | 80.2 | 81.6 | 87.4 | 72.1 | 82.3 | 82.2 | 82.9 |
| Ferret (You et al., 2023) | Vicuna-7B | 87.5 | 91.4 | 82.5 | 80.8 | 87.4 | 73.1 | 83.9 | 84.8 | 83.9 |
| Qwen-VL (Bai et al., 2023) | Qwen-7B | 88.6 | 92.3 | 84.5 | 82.8 | 88.6 | 76.8 | 86.0 | 86.3 | 85.7 |
| InternVL2-8B (Chen et al., 2024) | InternLM2.5-7B | 87.1 | 91.1 | 80.7 | 79.8 | 87.9 | 71.4 | 82.7 | 82.7 | 82.9 |
| LISA (Lai et al., 2024) | Vicuna-7B | 85.4 | 88.8 | 82.6 | 74.2 | 79.5 | 68.4 | 79.3 | 80.4 | 79.8 |
| GSVA (Xia et al., 2024) | Vicuna-7B | 86.3 | 89.2 | 83.8 | 72.8 | 78.8 | 68.0 | 81.6 | 81.8 | 80.3 |
| NEXT-Chat (Zhang et al., 2023) | Vicuna-7B | 85.5 | 90.0 | 77.9 | 77.2 | 84.5 | 68.0 | 80.1 | 79.8 | 80.4 |
| PixelLM (Ren et al., 2024) | Vicuna-7B | 89.8 | 92.2 | 86.4 | 83.2 | 87.0 | 78.9 | 84.6 | 86.0 | 86.0 |
| Groma (Ma et al., 2024) | Vicuna-7B | 89.5 | 92.1 | 86.3 | 83.9 | 88.9 | 78.1 | **86.4** | **87.0** | 86.5 |
| Text4Seg $_{\text{DeepseekVL-1.3B}}$ | DeepSeek-1.3B | 86.4 | 90.3 | 81.7 | 80.5 | 86.3 | 72.3 | 82.4 | 82.7 | 82.8 |
| Text4Seg $_{\text{DeepseekVL-7B}}$* | DeepSeek-7B | 87.2 | 90.8 | 83.4 | 82.1 | 88.1 | 76.8 | 81.1 | 81.0 | 83.8 |
| Text4Seg $_{\text{DeepseekVL-7B}}$ | DeepSeek-7B | 89.6 | 93.3 | 85.4 | 84.2 | **90.2** | 78.5 | 84.4 | 84.7 | 86.3 |
| Text4Seg $_{\text{Qwen-VL-7B}}$* | Qwen-7B | 87.2 | 90.1 | 83.6 | 82.1 | 87.4 | 76.6 | 81.5 | 81.3 | 83.7 |
| Text4Seg $_{\text{Qwen-VL-7B}}$ | Qwen-7B | 89.7 | 93.0 | 85.8 | 84.6 | 90.1 | 78.6 | 85.0 | 85.1 | 86.5 |
| Text4Seg $_{\text{LLaVA-1.5-7B}}$* | Vicuna-7B | 89.2 | 92.0 | 86.4 | 83.4 | 88.6 | 78.0 | 81.7 | 82.4 | 85.2 |
| Text4Seg $_{\text{LLaVA-1.5-7B}}$ | Vicuna-7B | **90.8** | **93.7** | **87.6** | 84.7 | **90.2** | 79.0 | 84.8 | 85.0 | 87.0 |
| Text4Seg $_{\text{InternVL2-8B}}$* | InternLM2.5-7B | 88.3 | 91.4 | 85.8 | 83.5 | 88.2 | 77.9 | 82.4 | 82.5 | 85.0 |
| Text4Seg $_{\text{InternVL2-8B}}$ | InternLM2.5-7B | 90.3 | 93.4 | 87.5 | **85.2** | 89.9 | **79.5** | 85.4 | 85.4 | **87.1** |
| *Generalist Segmentation Models (13B)* | | | | | | | | | | |
| Shikra (Chen et al., 2023a) | Vicuna-13B | 87.8 | 91.1 | 81.8 | 82.9 | 87.8 | 74.4 | 82.6 | 83.2 | 84.0 |
| LISA (Lai et al., 2024) | Vicuna-13B | 85.9 | 89.1 | 83.2 | 74.9 | 81.1 | 68.9 | 80.1 | 81.5 | 80.6 |
| GSVA (Xia et al., 2024) | Vicuna-13B | 87.7 | 90.5 | 84.6 | 76.5 | 81.7 | 70.4 | 83.9 | 84.9 | 82.5 |
| Text4Seg $_{\text{LLaVA-1.5-13B}}$* | Vicuna-13B | 89.6 | 92.3 | 87.0 | 84.4 | 89.0 | 79.1 | 82.9 | 82.9 | 85.9 |
| Text4Seg $_{\text{LLaVA-1.5-13B}}$ | Vicuna-13B | **91.2** | **94.3** | **88.0** | **85.7** | **90.8** | **80.1** | **85.6** | **85.5** | **87.7** |

the referring segmentation dataset by combining the `train` split of refCLEF, refCOCO, refCOCO+ (Kazemzadeh et al., 2014), and refCOCOg (Mao et al., 2016), resulting in a dataset of 800k samples. Our model is trained on this dataset for 5 epochs. Additionally, to evaluate the performance on a multi-object/non-object segmentation task, we construct a generalized referring expression segmentation dataset with 419k samples using the `train` split of grefCOCO (Liu et al., 2023a). We continue to fine-tune the model for 2 epochs.

**Result of single object.**  As summarized in Tab. 1, our Text4Seg achieves the highest performance across all splits of the refCOCO (+/g) datasets. For 7B-scale MLLMs, Text4Seg $_{\text{DeepseekVL-7B}}$ delivers an impressive average cIoU of 75.0, surpassing the closest competitor, Groundhog, which scores 74.2 cIoU. Notably, Text4Seg $_{\text{InternVL2-8B}}$ stands out with an average of 75.4 cIoU. At the 13B parameter scale, Text4Seg $_{\text{LLaVA-1.5-13B}}$ achieves a marked improvement, with an average cIoU of 76.2, significantly outperforming GSVA's 72.8 cIoU. Even without using the SAM refiner, our method remains competitive. For instance, Text4Seg $_{\text{InternVL2-8B}}^{\dagger}$, refined with CRFs, and Text4Seg $_{\text{InternVL2-8B}}^{\ddagger}$, based on $32 \times 32$ semantic descriptors, achieve results that rival or exceed existing methods.

**Result of multi-/no object.**  As shown in Tab. 2, Text4Seg maintains its competitive edge in multi-object and no-object referring expression segmentation tasks. For instance, at the 7B scale, Text4Seg records average scores between 69.9 and 71.1, a notable improvement over GSVA's 65.6 on the gRefCOCO dataset. At the 13B scale, Text4Seg $_{\text{LLaVA-1.5-13B}}$ further extends its lead, achieving an average score of 71.5, outperforming GSVA by 4.9 points. These outcomes highlight the robustness and versatility of Text4Seg in handling more complex segmentation challenges.

## 4.3 REFERRING EXPRESSION COMPREHENSION

**Settings.**  Our Text4Seg can also be directly applied in object detection with a simple *mask2box* paradigm, which first generates a segmentation mask based on the input and then derives the bounding box from the mask. We employ this method to evaluate the referring expression comprehension of our model using the same datasets as in RES. Specifically, a prediction is considered correct if the IoU between the predicted and ground truth bounding boxes exceeds 0.5.

Table 4: Results on **visual question answering** and **RES** benchmarks. refC denotes refCOCO. Mix[†] is a combination of referring segmentation, semantic segmentation and VQA datasets from LISA.

| Methods | Training Data | VQA | | | | | | RES (val) | | |
|---------|---------------|------|-----|--------|-----------|--------|------|------|-------|-------|
| | | VQAv2 | GQA | VisWiz | ScienceQA | TextQA | POPE | refC | refC+ | refCg |
| LISA | Mix[†] | - | - | - | - | - | - | 74.1 | 62.4 | 66.4 |
| LLaVA-1.5 | 665k | 78.0 | 61.7 | 50.6 | 68.4 | 55.0 | 85.4 | - | - | - |
| Text4Seg | 665k + refseg | 76.6 | 60.2 | 50.9 | 68.1 | 55.0 | 84.2 | 77.5 | 70.7 | 73.4 |

**Results.** As shown in Tab. 3, our Text4Seg achieves the best results on the refCOCO and refCOCO+ datasets, while Groma performs well on refCOCOg. However, Text4Seg $_{\text{InternVL2-8B}}$ delivers the highest overall accuracy, reaching 87.1%. Notably, both Text4Seg $_{\text{InternVL2-8B}}$ and Text4Seg $_{\text{Qwen-VL-7B}}$ surpass their respective MLLM baselines. In particular, Text4Seg $_{\text{InternVL2-8B}}$ demonstrates a significant improvement over InternVL2-8B, increasing its average accuracy from 82.9% to 87.1%. Additionally, our Text4Seg $_{\text{LLaVA-1.5-13B}}$ outperforms previous SOTA, Shikra, by an average margin of 3.7%. It is worth noting that Text4Seg $_{\text{LLaVA-1.5-7B}}$* and Text4Seg $_{\text{LLaVA-1.5-13B}}$*, without a mask refiner, outperform their respective baseline counterparts. These results emphasize the superiority of Text4Seg in following instructions, leading to enhanced visual grounding ability.

## 4.4 VISUAL UNDERSTANDING

**Settings.** Our text-as-mask paradigm allows for seamless integration of downstream segmentation task into the pre-training of MLLMs. To evaluate its effectiveness, we assess the model's performance on various visual understanding benchmarks, using the LLaVA-1.5-7B model as the baseline. Our method, Text4Seg, built upon the stage-2 of LLaVA-1.5-7B, is trained on both the LLaVA-v1.5-mix665k dataset and our referring segmentation datasets. For a comprehensive comparison, we also report the performance of the LLaVA-1.5-7B model based on our implementation.

**Results.** Table 4 presents a comparison between LLaVA-1.5 and Text4Seg across various VQA and RES benchmarks. Notably, Text4Seg, trained on a mixed dataset, achieves performance on par with LLaVA-1.5 in visual question answering tasks while delivering strong results in RES benchmarks. These results validate that our text generation based segmentation method acts as a seamless enhancement, offering a streamlined approach for pre-training MLLMs. It successfully integrates robust segmentation functionality without compromising the model's conversational capabilities.

## 4.5 OPEN VOCABULARY SEGMENTATION

**Settings.** We follow LaSagnA (Wei et al., 2024) to evaluate the performance of Text4Seg on open-vocabulary segmentation tasks. Our Text4Seg is built upon LLaVA-1.5-7B and trained on the COCOStuff (Caesar et al., 2018) for 1 epoch. We evaluate the model's performance on ADE20K (A-150) (Zhou et al., 2019), PASCAL Context 59 (PC-59) (Mottaghi et al., 2014), and PASCAL VOC 20 (PAS-20) (Everingham, 2009) datasets, using mIoU as the evaluation metric.

Table 5: **Open Vocabulary Segmentation** results (mIoU) on various segmentation datasets.

| Methods | A-150 | PC-59 | PAS-20 |
|---------|-------|-------|--------|
| *Specialised Segmentation Models* | | | |
| ClearCLIP | 16.7 | 35.9 | 80.9 |
| ProxyCLIP | 24.2 | 39.6 | 83.3 |
| MaskCLIP | 23.7 | 45.9 | - |
| GroupViT | 9.2 | 23.4 | 79.7 |
| OVSeg | 24.8 | 53.3 | 92.6 |
| SAN | 27.5 | 53.8 | 94.0 |
| *Generalist Segmentation Models (7B)* | | | |
| LaSagnA | 14.3 | 46.1 | 69.8 |
| Text4Seg | **16.5** | **52.5** | **76.5** |

**Results.** As reported in the Tab. 5, it is expected that Text4Seg falls behind specialized segmentation models (*e.g.*, ClearCLIP (Lan et al., 2024a), ProxyCLIP (Lan et al., 2024b), MaskCLIP (Ding et al., 2022), GroupViT (Xu et al., 2022), OVSeg (Liang et al., 2023), and SAN (Xu et al., 2023)), because LLMs typically require quite large datasets to be sufficiently trained. However, Text4Seg still demonstrates competitive performance on the PC-59 benchmark, underscoring its efficiency. More importantly, it significantly outperforms the MLLM-based LaSagnA, which uses an additional decoder, showcasing its strong potential for open-vocabulary segmentation.

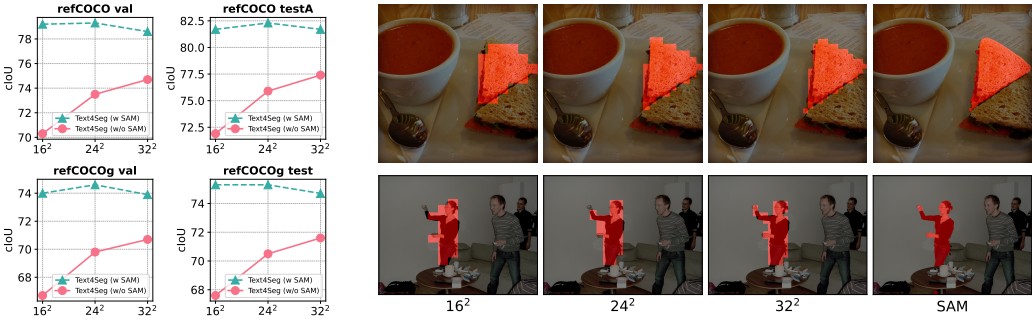

Figure 6: RES comparison across different resolutions.

Figure 7: Visualization of RES results across different resolutions, and with SAM as mask refiner.

Table 6: Ablation study of mask refiner on refCOCO val.

| Method | Refiner | cIoU | Acc@0.5 | Time (s) |
|--------|---------|------|---------|----------|
| Text4Seg | None | 73.5 | 89.3 | 5.34 |
| Text4Seg | SAM-B | 75.5 | 89.9 | 5.54 |
| Text4Seg | SAM-L | 79.1 | 90.6 | 5.73 |
| Text4Seg | SAM-H | 79.2 | 90.0 | 5.92 |

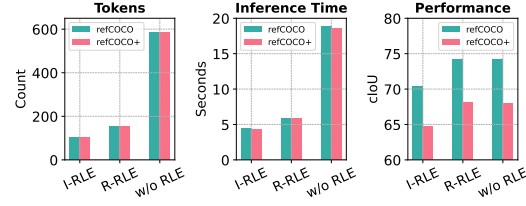

Figure 8: R-RLE is better than I-RLE.

### 4.6 ABLATION STUDY

Focusing on semantic descriptors for visual segmentation and grounding, we conducted ablation studies to evaluate its impact on performance using InternVL2-8B (Chen et al., 2024) as the MLLM.

**Resolution of semantic descriptors.** To analyze the impact of varying the resolution of semantic descriptors on RES performance, we create instruction-tuning datasets with different densities of semantic descriptors. Specifically, we represent each image with $16\times16$, $24\times24$, and $32\times32$ semantic descriptors to explore how finer or coarser resolutions affect model accuracy. As shown in Fig. 6, the performance of Text4Seg without a mask refiner improves with higher resolution, from 67.5 cIoU at $16^2$ to 71.4 cIoU at $32^2$ on average, surpassing LISA at 69.9 cIoU. Two examples are illustrated in Fig. 7. *Note that the improvement is achieved without increasing the feature resolution from the vision tower of MLLM.* While higher-density semantic descriptors improve results, it also significantly increases token length and computational cost. Therefore, we incorporate an off-the-shelf SAM to refine the outputs. Experimental results show that using $16^2$ semantic descriptors with SAM already achieves optimal performance.

**Mask refiner with SAM variants.** Tab. 6 compares the performance of various mask refiners, such as SAM with different architectures, against no refiner for semantic descriptors at a $16 \times 16$ resolution. SAM with the ViT-L architecture achieves similar performance to SAM with ViT-H while reducing inference time. Notably, Text4Seg with SAM-L increases the average performance on RES tasks from 73.5 to 79.1 cIoU compared to Text4Seg without a mask refiner, with only a little increase in inference time.

**I-RLE v.s. R-RLE.** We investigate the impact of different encoding methods for semantic descriptors at a $16 \times 16$ resolution using the train/val splits of the refCOCO and refCOCO+ datasets. As illustrated in Fig. 8, while full-length semantic descriptors achieve high performance, they suffer from significantly longer inference times (∼19 seconds) due to longer output tokens (∼590) on both datasets. Although the I-RLE method reduces both the number of tokens and inference time, it results in a notable performance drop, from 74.2 to 70.4 cIoU on refCOCO and 68.0 to 64.7 cIoU on refCOCO+. Our proposed R-RLE method strikes a better balance, reducing the length of semantic descriptors by 74% and improving inference speed by an average of $3\times$, while still maintaining the same performance.

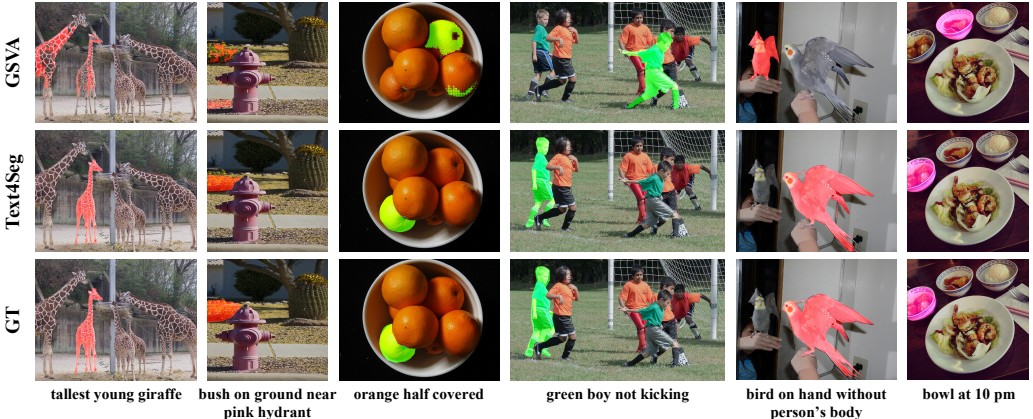

Figure 9: Visualizations of Text4Seg and GSVA (Xia et al., 2024) on the RES task. Our Text4Seg is based on InternVL2 backbone. The corresponding referring expressions are displayed in the bottom.

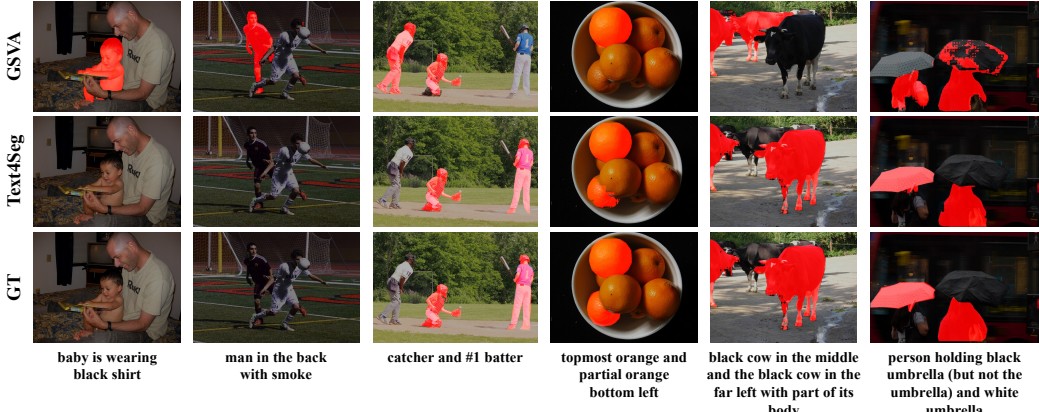

Figure 10: Visualizations of Text4Seg and GSVA (Xia et al., 2024) on the GRES task.

### 4.7 VISUALIZATION EXAMPLES

We present qualitative comparisons between Text4Seg and GSVA in Figs. 9 and 10. In the single-object RES task, Text4Seg demonstrates a superior understanding of referring expressions, generating more accurate and precise segmentation maps compared to GSVA. In the GRES task (Fig. 10), GSVA tends to incorrectly segment empty objects despite the inclusion of a <REJ> token (as seen in the first two columns). In contrast, Text4Seg consistently avoids such mistakes by labeling them as "others" without special design. Furthermore, Text4Seg significantly outperforms GSVA in the multiple-object RES task, delivering more precise segmentation results with better grounding performance. These results fully validate the effectiveness of Text4Seg in handling diverse and challenging visual grounding and segmentation tasks.

## 5 CONCLUSION

In this work, we present Text4Seg, a decoder-free framework that integrates seamlessly with existing MLLMs for image segmentation using a novel *text-as-mask* paradigm. With the novel semantic descriptors, Text4Seg achieves state-of-the-art performance across various segmentation tasks, without requiring architecture modifications. We further introduce the Row-wise Run-Length Encoding (R-RLE) to compress semantic descriptors, which significantly improves the efficiency of Text4Seg while maintaining the performance. In summary, this work highlights the flexibility and effectiveness of Text4Seg in bridging the gap between MLLMs and vision-centric tasks, offering a scalable solution for future research in multimodal learning.

**Acknowledgment.** This study is supported under the RIE2020 Industry Alignment Fund – Industry Collaboration Projects (IAF-ICP) Funding Initiative, as well as cash and in-kind contribution from the industry partner(s).

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

## A    ADDITIONAL IMPLEMENTATION DETAILS

### A.1    IMPLEMENTATION OF ADOPTING SAM AS MASK REFINER.

We employ SAM with a ViT-H architecture as our mask refiner. For referring expression segmentation tasks, we refine the coarse masks produced by Text4Seg from the semantic descriptors using the following process:

- **Step 1:** Convert the binary mask into a logit representation by applying the inverse sigmoid function.
- **Step 2:** Randomly select 10 positive and 10 negative points from the coarse binary mask.
- **Step 3:** Provide the selected points as point prompts, the logit representation as a mask prompt, and the RGB image as input to SAM, generating a refined mask and updated logits.
- **Step 4:** Repeat Step 3 twice.

This iterative process helps enhance the quality of the segmentation mask. The final mask produced by SAM is then resized to the original image dimensions, resulting in pixel-level segmentation masks. For open-vocabulary segmentation, this strategy is applied iteratively across multiple class masks, which are then combined to form the final segmentation maps.

### A.2    DETAILS OF TRAINING HYPER-PARAMETERS

Table 7 presents the training hyperparameters used for training Text4Seg on the referring expression segmentation task. We primarily adhere to the same settings as LLaVA-1.5, and these parameters are consistently applied across other tasks as well.

Table 7: Hyper-parameters and training settings for RES task.

|  | Param Name | Value |
|---|---|---|
| Optimizer | Type | AdamW |
|  | Learning rate | 2e-4 |
|  | Weight decay | 0.0 |
|  | $(\beta_1, \beta_2)$ | (0.9, 0.95) |
|  | Gradient norm clip | 1.0 |
|  | Scheduler | Linearly decay |
|  | Warmup ratio | 0.03 |
| LoRA | Rank | 64 |
|  | Alpha $(\alpha)$ | 128 |
|  | Dropout | 0.05 |
|  | Module | Linear layers of connector and LLMs |
| Training | Trainable #Params. | About 2% of the LLM (7B $\rightarrow$ 160M) |
|  | Numerical precision | FP16 |
|  | Global batch size | 128 |
|  | Number of samples per epoch | 800k |
|  | Total epochs | 5 |
|  | GPUs | A800(40G) $\times$ 8 |
|  | Time | About 2 Days |

## B    COMPARISON OF TRAINING DATASETS

Most prior methods follow a two-stage training paradigm: **Continued Pre-Training (CPT)** using large datasets, followed by **Supervised Fine-Tuning (SFT)** for specific tasks. The datasets used in these approaches are summarized in the following tables:

- Tab. 8: Datasets for **Continued Pre-Training (CPT)**

- Tab. 9: Datasets for **Supervised Fine-Tuning (SFT)** in **Referring Expression Segmentation (RES)**
- Tab. 10: Datasets for **Supervised Fine-Tuning (SFT)** in **Generalized Referring Expression Segmentation (GRES)**

We can note that:

1. For CPT, previous methods rely heavily on large and diverse datasets, whereas our approach, Text4Seg, eliminates this requirement, demonstrating superior efficiency and effectiveness.

2. For SFT, we ensure a fair comparison by following previous works and train on:
   - The `train` split of refCOCO series for **RES** and **REC** tasks.
   - The `train` split of grefCOCO for the **GRES** task.

Table 8: Training datasets of **Continued Pre-Training (CPT)**.

| Methods | Datasets |
|---|---|
| LISA | ADE20K, COCO-Stuff, PACO-LVIS, PartImageNet, PASCAL-Part, refCLEF, refCOCO, refCOCO+, refCOCOg, LLaVA-v1.5-mix665k |
| PixelLM | ADE20K, COCO-Stuff, PACO-LVIS, refCLEF, refCOCO, refCOCO+, refCOCOg, LLAVA-150k, multi-target reasoning segmentation (MUSE) |
| GSVA | ADE20K, COCO-Stuff, PACO-LVIS, Mapillary Vistas, PASCAL-Part, refCLEF, refCOCO, refCOCO+, refCOCOg, gRefCOCO, LLaVA-Instruct-150K, ReasonSeg |
| AnyRef | ADE20K, COCO-Stuff, PACO-LVIS, refCLEF, refCOCO, refCOCO+, refCOCOg, PhraseCut, Flickr30K Entities, AVSBench |
| NEXT-Chat | Flickr30K Entities, Visual Genome, RefCOCO, RefCOCO+, RefCOCOg, VQAv2, PointQA, Visual7W, VCR, LLaVA-Instruct-150K, VG grounded captioning, Shikra-RD |
| Groundhog | Multi-Modal Multi-Grained Grounding dataset (M3G2): PNG, Flickr30K-Entity, refCLEF, refCOCO, refCOCO+, refCOCOg, gRefCOCO, PhraseCut, D-Cube, ReasonSeg, RIO, SK-VG, VizWiz-G, TextVQA-X, GQA, VQS, Shikra-BinaryQA, EntityCount, FoodSeg-QA, LVIS-QA, RefCOCO-REG, RefCOCO+-REG, RefCOCOg-REG, gRefCOCO-REG, VG-SpotCap, V7W, PointQA, VCR, ShikraRD, SVIT-RD, Guesswhat, VG-RefMatch, HierText |
| GLaMM | Grounding-anything Dataset (GranD): **11M** images, **810M** masks, **84M** referring expressions, GranD-f |
| Text4Seg | **None** |

Table 9: Referring Expression Segmentation Datasets of **Supervised Fine-Tuning (SFT)**. [†] Other methods have already incorporated refCLEF dataset in their CPT training datasets.

| Methods | Datasets |
|---|---|
| LISA | refCOCO, refCOCO+, refCOCOg |
| PixelLM | None |
| GSVA | refCOCO, refCOCO+, refCOCOg |
| AnyRef | refCOCO, refCOCO+, refCOCOg |
| NEXT-Chat | refCOCO, refCOCO+, refCOCOg |
| Groundhog | None |
| GLaMM | refCOCO, refCOCO+, refCOCOg |
| Text4Seg | refCOCO, refCOCO+, refCOCOg, refCLEF[†] |

## C  ADDITIONAL VISUAL INSTRUCTION DATA DETAILS

**Query-answer template.**  We provide the question-answer templates in the Figs. 11 to 13. For partial segmentation tasks, the templates are designed to segment **only a subset of objects in the**

Table 10: Generalized Referring Expression Segmentation Datasets of **Supervised Fine-Tuning (SFT)**.

| Methods | Datasets |
|---------|----------|
| LISA | grefCOCO |
| GSVA | grefCOCO |
| Text4Seg | grefCOCO |

**image**, such as a single object in the RES task, multiple objects in the GRES task, or partial labels in semantic segmentation tasks. For conditioned segmentation tasks, the user provides a list of condition labels, and the model segments the entire image based on those specified labels. For open-vocabulary segmentation tasks, the model leverages its open-vocabulary capabilities to segment the image and label all detected categories.

**Visual instruction data on RES datasets.** We adopt the question-answer templates from Fig. 11 to construct the training data. Specifically, we iterate through all `<image, referring expression, mask>` pairs in the dataset, transforming the vanilla mask into semantic descriptors, using the referring expression as the descriptor. The referring expression is placed in the `[class_name]` placeholder within each question-answer template. The RES training set is constructed by combining the `train` splits of refCLEF, refCOCO, refCOCO+, and refCOCOg, with the process repeated twice. This results in a final RES training set comprising 800k samples. The same method is applied to construct the GRES training set, which contains 419k samples.

**Visual instruction data on open-vocabulary segmentation datasets.** For the open-vocabulary segmentation task, we utilize all three types of question-answer templates. Specifically, we construct our visual instruction data using the COCOStuff dataset. The ratio of open-vocabulary segmentation templates, partial segmentation templates, and conditioned segmentation templates is set to $1 : 3 : 6$. To further enhance diversity, we apply random cropping to both the image and mask. By iterating 10 times over the COCOStuff `train` set, we ultimately generate a training dataset consisting of 1.16M samples.

## D ADDITIONAL QUANTITATIVE RESULTS

### D.1 MORE RESULTS ON MASK REFINER

We present additional ablation study results on the mask refiner in Tab. 11, evaluated on the `val` split of the refCOCO(+/g) datasets. The findings indicate that both SAM with ViT-L and ViT-H architectures achieve similarly strong performance across all datasets, demonstrating the robustness of the mask refinement process regardless of the test datasets.

Table 11: Ablation study on mask refiner on refCOCO (+/g) datasets.

| Method | Refiner | refCOCO val | | | refCOCO+ val | | | refCOCOg val | | |
|--------|---------|------|---------|----------|------|---------|----------|------|---------|----------|
| | | cIoU | Acc@0.5 | Time (s) | cIoU | Acc@0.5 | Time (s) | cIoU | Acc@0.5 | Time (s) |
| Text4Seg | None | 73.5 | 89.3 | 5.34 | 67.6 | 83.6 | 5.26 | 69.8 | 84.0 | 6.18 |
| Text4Seg | SAM-B | 75.5 | 89.9 | 5.54 | 69.8 | 84.7 | 5.46 | 71.3 | 84.6 | 6.30 |
| Text4Seg | SAM-L | 79.1 | 90.6 | 5.73 | 72.8 | 85.1 | 5.63 | 74.2 | 85.2 | 6.58 |
| Text4Seg | SAM-H | 79.3 | 90.0 | 5.92 | 72.6 | 84.3 | 5.84 | 74.6 | 85.6 | 6.75 |

### D.2 MORE RESULTS ON DIFFERENT RESOLUTION OF SEMANTIC DESCRIPTORS

Figure 14 provides the complete results across all RES datasets, including refCOCO+. The results indicate that using a $16 \times 16$ length of semantic descriptors, combined with the SAM refiner, is an effective approach that delivers strong performance. While it is possible to eliminate the SAM refiner by further increasing the density of semantic descriptors, this would demand significantly higher computational resources, and we will leave this optimization for future work.

---

**Question:**

○ "Please segment only the [class_name] in the image.",
○ "Can you segment the [class_name] in the image?",
○ "Where is the [class_name] in this picture? Please respond with segmentation mask.",
○ "Where is '[class_name]' in this image? Please output segmentation mask.",
○ "Could you provide the segmentation mask for '[class_name]' in this image?",
○ "Please segment the image and highlight '[class_name]'."

**Answer:**

• "Sure, here is the segmentation mask for '[class_name]':",
• "Here is the segmentation map focusing on the [class_name]:",
• "Here is the segmentation mask highlighting the [class_name]:",
• "The segmentation map for '[class_name]' is:",
• "The segmentation mask for '[class_name]' is shown below:",
• "Sure, Here's the segmentation of the [class_name]:",
• "Sure, the segmented output for '[class_name]' is:",
• "Certainly, the segmentation map for '[class_name]' is:",
• "Certainly, here is the segmentation mask for '[class_name]':",
• "The segmentation mask for '[class_name]' is shown below:"

---

Figure 11: Question-Answer-Template for **partial segmentation** tasks, such as referring segmentation and open vocabulary segmentation tasks. [class_name] will be replace with the referring expression in RES datasets or the selected class list in semantic segmentation datasets. The semantic descriptors are appended at the end of each answer.

### D.3   MORE RESULTS REGARDING THE MASK REFINER

We provide additional quantitative results on Tabs. 12 to 14. While Text4Seg without a mask refiner slightly lags behind LISA and GSVA in terms of average cIoU on referring expression segmentation (RES) tasks, traditional mask refinement techniques, such as Conditional Random Fields (CRF), can be employed to enhance segmentation accuracy. For instance, Text4Seg $_{\text{InternVL2-8B}}$ with a CRF refiner improves the baseline performance from an average cIoU of 67.5 to 70.1 on RES tasks. Additionally, when using $32 \times 32$ semantic descriptors, Text4Seg outperforms its counterpart with $16 \times 16$ descriptors. Specifically, Text4Seg $_{\text{InternVL2-8B}}$ with $32 \times 32$ semantic descriptors achieves an average cIoU of 71.4, surpassing LISA's 69.9 and matching GSVA's 71.4 on RES tasks. On the GRES tasks, as shown in the Tab. 13, both CRF and SAM refiners significantly enhance performance, outperforming LISA and GSVA. Notably, Text4Seg $_{\text{InternVL2-8B}}$ with $32 \times 32$ semantic descriptors, even without a mask refiner, achieves performance superior to existing methods. Finally, on the REC tasks, Text4Seg without a SAM refiner continues to outperform current methods, further demonstrating the effectiveness of Text4Seg's visual grounding capabilities.

## E   ADDITIONAL QUALITATIVE RESULTS

In this section, we provide more visual examples for different tasks to show the strong capabilities of the proposed Text4Seg.

**Referring expression segmentation.**   Figure 15 provides additional examples of Text4Seg applied to the referring expression segmentation (RES) task. It is evident that Text4Seg can segment objects based on various criteria, including different classes (*e.g.*, "clear glass"), colors (*e.g.*, "blue"), and positions (*e.g.*, "food in the back right"). This versatility demonstrates its superiority in accurately identifying and segmenting objects in complex scenarios.

**Question:**

○ "Please segment the image based on the category: [class_name].",
○ "Segment the image according to the specified category: [class_name].",
○ "Segment the image while focusing on the category: [class_name].",
○ "Please provide a segmentation map for the category: [class_name].",
○ "Segment the image with emphasis on the class: [class_name].",
○ "Please segment the image, focusing on the candidate category: [class_name].",
○ "Could you segment the image, considering the indicated class: [class_name]?"

**Answer:**

• "Sure, here is the segmentation based on the category '[class_name]':",
• "The image has been segmented according to the category '[class_name]':",
• "Certainly, here is the segmentation map for the category '[class_name]':",
• "The image is segmented with emphasis on the class '[class_name]':",
• "Here is the segmented image focusing on the candidate category '[class_name]':",
• "The image has been segmented with the category '[class_name]' in mind:",
• "Sure, the segmentation mask is:",
• "Sure, the segmented image is:",
• "Certainly, the segmented map is:",
• "Certainly, here is the segmentation mask:",
• "Certainly, here is the segmented output:",
• "Sure, here is the segmentation map:",
• "The segmentation mask is shown below:"

Figure 12: Question-Answer-Template for **conditioned segmentation** tasks like open vocabulary segmentation task. [class_name] will be replace with the condition class list in semantic segmentation datasets. The semantic descriptors are appended at the end of each answer.

**Referring expression comprehension.** We also present additional results on the Referring Expression Comprehension (REC) task in Fig. 16. It is evident that the coarse masks generated by Text4Seg can be effectively utilized for object localization tasks using the simple *mask2box* method. This application highlights the accuracy of Text4Seg in referring object localization, demonstrating its capability to precisely identify and locate objects within complex images.

**Open vocabulary semantic segmentation.** Figure 17 presents additional examples of Text4Seg performing open-vocabulary segmentation. Notably, Text4Seg demonstrates its ability to segment not only common large objects but also small objects effectively, such as the person and boat on the river. This versatility highlights Text4Seg's proficiency in accurately identifying and segmenting a wide range of object sizes. Figure 18 illustrates the multi-object segmentation capabilities of Text4Seg. It is evident that Text4Seg successfully segments all identified objects within the image, showcasing its strong ability to handle multiple objects in complex scenarios. This performance highlights its robustness and effectiveness in accurately distinguishing various elements within a single scene.

**Visual understanding.** Figure 19 presents an example where Text4Seg is used for image captioning, single-object segmentation, and multi-object segmentation. Additionally, Fig. 20 compares the image reasoning capabilities of Text4Seg with the original LLaVA-1.5. While maintaining similar reasoning abilities, our proposed Text4Seg extends functionality by enabling segmentation tasks.

**Question:**

o "Segment the entire image and classify each category separately."
o "Please perform segmentation on this image and highlight all identifiable elements."
o "Perform segmentation on this image and label all detected categories."
o "Please identify and segment all categories present in the image."
o "Segment the image and label all categories detected."
o "Could you segment the image and label each identifiable category?"
o "Segment the image to identify and label all visible categories."
o "Segment and classify all elements in the image."
o "Identify and segment all categories visible in the image."
o "Can you segment and label the image?"
o "Might you segment this image?"
o "Can you perform segmentation on this image?"
o "Could you please segment this image?"

**Answer:**

• "Sure, here is the segmented image with each category classified separately:"
• "Sure, here's the segmented image showing all visible categories:"
• "The image is segmented and annotated with each category:"
• "The image segmentation is complete, with all categories marked:"
• "Sure, the segmentation mask is:"
• "Sure, the segmented image is:"
• "Certainly, the segmented map is:"
• "Certainly, here is the segmentation mask:"
• "Certainly, here is the segmented output:"
• "Sure, here is the segmentation map:"
• "The segmentation mask is shown below:"

Figure 13: Question-Answer-Template for **open vocabulary segmentation** tasks. Following LaSagnA (Wei et al., 2024), the class label lists of the test benchmarks are given in the question for fair quantitative comparison. The semantic descriptors are appended at the end of each answer.

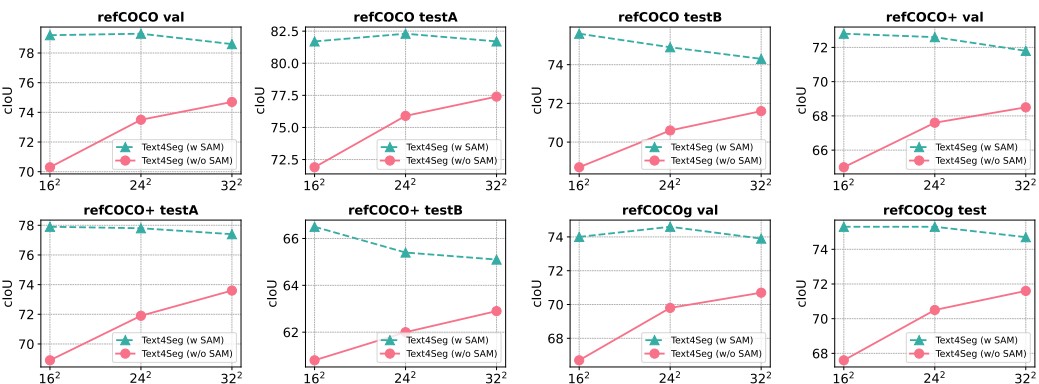

Figure 14: Text4Seg with different resolutions of semantic descriptors on all RES datasets.

Table 12: **Additional Referring Expression Segmentation** results (cIoU) on refCOCO (+/g) datasets. ‡ Model is based on the semantic descriptors with a resolution of 32×32.

| Methods | Refiner | refCOCO | | | refCOCO+ | | | refCOCOg | | Avg. |
|---|---|---|---|---|---|---|---|---|---|---|
| | | val | testA | testB | val | testA | testB | val | test | |
| *Generalist Segmentation Models (≤8B)* | | | | | | | | | | |
| LISA (Lai et al., 2024) | - | 74.9 | 79.1 | 72.3 | 65.1 | 70.8 | 58.1 | 67.9 | 70.6 | 69.9 |
| GSVA (Xia et al., 2024) | - | 77.2 | 78.9 | 73.5 | 65.9 | 69.6 | 59.8 | 72.7 | 73.3 | 71.4 |
| Text4Seg $_{\text{DeepseekVL-1.3B}}$ | None | 66.2 | 68.7 | 63.6 | 60.7 | 64.5 | 54.9 | 64.2 | 64.2 | 63.4 |
| Text4Seg $_{\text{DeepseekVL-1.3B}}$ | SAM-H | 75.0 | 78.6 | 70.1 | 68.4 | 73.4 | 60.0 | 71.5 | 71.7 | 71.1 |
| Text4Seg $_{\text{DeepseekVL-7B}}$ | None | 69.7 | 71.2 | 67.9 | 64.5 | 68.0 | 60.2 | 66.6 | 66.7 | 66.9 |
| Text4Seg $_{\text{DeepseekVL-7B}}$ | CRF | 72.6 | 74.8 | 70.0 | 67.2 | 71.5 | 62.2 | 69.1 | 69.4 | 69.6 |
| Text4Seg $_{\text{DeepseekVL-7B}}$ | SAM-H | 78.8 | 81.5 | 74.9 | 72.5 | 77.4 | 65.9 | 74.3 | 74.4 | 75.0 |
| Text4Seg $_{\text{Qwen-VL-7B}}$ | None | 68.3 | 70.0 | 67.3 | 63.1 | 67.2 | 59.9 | 66.5 | 66.4 | 66.1 |
| Text4Seg $_{\text{Qwen-VL-7B}}$ | CRF | 71.3 | 73.7 | 69.6 | 65.9 | 70.4 | 61.9 | 69.3 | 69.3 | 68.9 |
| Text4Seg $_{\text{Qwen-VL-7B}}$ | SAM-H | 78.0 | 80.9 | 74.6 | 71.6 | 77.3 | 66.0 | **74.8** | 74.7 | 74.7 |
| Text4Seg $_{\text{LLaVA-1.5-7B}}$ | None | 70.5 | 72.3 | 69.3 | 64.4 | 68.7 | 60.6 | 65.1 | 66.5 | 67.2 |
| Text4Seg $_{\text{LLaVA-1.5-7B}}$ | CRF | 73.2 | 75.7 | 71.4 | 67.0 | 71.9 | 62.4 | 67.3 | 68.9 | 69.7 |
| Text4Seg $_{\text{LLaVA-1.5-7B}}$ | SAM-H | **79.3** | **81.9** | **76.2** | 72.1 | 77.6 | 66.1 | 72.1 | 73.9 | 74.9 |
| Text4Seg $_{\text{InternVL2-8B}}$ | None | 70.3 | 71.9 | 68.7 | 65.0 | 68.9 | 60.8 | 66.7 | 67.6 | 67.5 |
| Text4Seg $_{\text{InternVL2-8B}}$ | CRF | 73.0 | 75.2 | 70.7 | 67.6 | 72.1 | 62.6 | 68.9 | 70.3 | 70.1 |
| Text4Seg $_{\text{InternVL2-8B}}$ | SAM-H | 79.2 | 81.7 | 75.6 | **72.8** | **77.9** | **66.5** | 74.0 | **75.3** | **75.4** |
| Text4Seg $_{\text{InternVL2-8B}}$ ‡ | None | 74.7 | 77.4 | 71.6 | 68.5 | 73.6 | 62.9 | 70.7 | 71.6 | 71.4 |
| Text4Seg $_{\text{InternVL2-8B}}$ ‡ | SAM-H | 78.6 | 81.7 | 74.3 | 71.8 | 77.4 | 65.1 | 73.9 | 74.7 | 74.7 |
| *Generalist Segmentation Models (13B)* | | | | | | | | | | |
| LISA (Lai et al., 2024) | - | 76.0 | 78.8 | 72.9 | 65.0 | 70.2 | 58.1 | 69.5 | 70.5 | 70.1 |
| GSVA (Xia et al., 2024) | - | 78.2 | 80.4 | 74.2 | 67.4 | 71.5 | 60.9 | **74.2** | **75.6** | 72.8 |
| Text4Seg $_{\text{LLaVA-1.5-13B}}$ | None | 71.3 | 72.9 | 70.3 | 65.9 | 70.0 | 61.8 | 66.8 | 67.6 | 68.3 |
| Text4Seg $_{\text{LLaVA-1.5-13B}}$ | CRF | 74.1 | 76.4 | 72.4 | 68.5 | 72.8 | 63.6 | 69.1 | 70.1 | 70.9 |
| Text4Seg $_{\text{LLaVA-1.5-13B}}$ | SAM-H | **80.2** | **82.7** | **77.3** | **73.7** | **78.6** | **67.6** | 74.0 | 75.1 | **76.2** |

Table 13: **Additional Generalized Referring Expression Segmentation** results on the grefCOCO dataset. [‡] Model is based on the semantic descriptors with a resolution of 32×32.

| Methods | Refiner | Validation Set | | Test Set A | | Test Set B | | Avg. |
|---|---|---|---|---|---|---|---|---|
| | | gIoU | cIoU | gIoU | cIoU | gIoU | cIoU | |
| *Generalist Segmentation Models (≤8B)* | | | | | | | | |
| LISA (Lai et al., 2024) | - | 61.6 | 61.8 | 66.3 | 68.5 | 58.8 | 60.6 | 62.9 |
| GSVA (Xia et al., 2024) | - | 66.5 | 63.3 | 71.1 | 69.9 | 62.2 | 60.5 | 65.6 |
| Text4Seg DeepseekVL-1.3B | None | 64.3 | 57.2 | 62.2 | 61.2 | 57.1 | 54.9 | 59.5 |
| Text4Seg DeepseekVL-1.3B | SAM-H | 69.9 | 63.2 | 69.7 | 67.5 | 62.3 | 59.8 | 65.4 |
| Text4Seg DeepseekVL-7B | None | 69.0 | 62.7 | 66.3 | 65.9 | 62.1 | 61.1 | 64.5 |
| Text4Seg DeepseekVL-7B | CRF | 70.4 | 65.8 | 68.9 | 69.9 | 63.2 | 63.6 | 67.0 |
| Text4Seg DeepseekVL-7B | SAM-H | **74.7** | 69.0 | 74.3 | 73.0 | 67.4 | 66.3 | 70.8 |
| Text4Seg Qwen-VL-7B | None | 68.5 | 61.1 | 64.6 | 63.6 | 61.1 | 59.6 | 63.1 |
| Text4Seg Qwen-VL-7B | CRF | 69.7 | 64.1 | 67.4 | 67.8 | 62.4 | 62.3 | 65.6 |
| Text4Seg Qwen-VL-7B | SAM-H | 74.4 | 68.1 | 73.1 | 71.5 | 66.7 | 65.3 | 69.9 |
| Text4Seg LLaVA-1.5-7B | None | 67.9 | 61.6 | 66.2 | 65.9 | 60.9 | 59.8 | 63.7 |
| Text4Seg LLaVA-1.5-7B | CRF | 69.1 | 64.7 | 69.9 | 70.8 | 62.1 | 62.3 | 66.5 |
| Text4Seg LLaVA-1.5-7B | SAM-H | 73.6 | 67.9 | 74.1 | 72.8 | 66.1 | 64.8 | 69.9 |
| Text4Seg InternVL2-8B | None | 68.8 | 63.1 | 66.9 | 67.1 | 62.1 | 61.6 | 64.9 |
| Text4Seg InternVL2-8B | CRF | 70.0 | 66.1 | 69.4 | 70.9 | 63.1 | 64.1 | 67.3 |
| Text4Seg InternVL2-8B | SAM-H | 74.4 | **69.1** | **75.1** | **73.8** | **67.3** | **66.6** | **71.1** |
| Text4Seg InternVL2-8B[‡] | None | 71.8 | 65.6 | 71.2 | 70.0 | 64.2 | 62.5 | 67.6 |
| Text4Seg InternVL2-8B[‡] | SAM-H | 74.9 | 68.8 | 75.4 | 73.6 | 67.0 | 65.1 | 70.8 |
| *Generalist Segmentation Models (13B)* | | | | | | | | |
| LISA (Lai et al., 2024) | - | 63.5 | 63.0 | 68.2 | 69.7 | 61.8 | 62.2 | 64.7 |
| GSVA (Xia et al., 2024) | - | 68.0 | 64.1 | 71.8 | 70.5 | 63.8 | 61.3 | 66.6 |
| Text4Seg LLaVA-1.5-13B | None | 69.2 | 63.9 | 67.4 | 67.6 | 62.7 | 62.0 | 65.5 |
| Text4Seg LLaVA-1.5-13B | CRF | 70.3 | 66.9 | 69.8 | 71.4 | 63.8 | 64.4 | 67.8 |
| Text4Seg LLaVA-1.5-13B | SAM-H | **74.8** | **69.8** | **75.1** | **74.3** | **68.0** | **67.1** | **71.5** |

Table 14: **Additional Referring Expression Comprehension** results (Acc@0.5) on RefCOCO (+/g) datasets. [‡] Model is based on the semantic descriptors with a resolution of 32×32.

| Methods | Refiner | refCOCO | | | refCOCO+ | | | refCOCOg | | Avg. |
|---|---|---|---|---|---|---|---|---|---|---|
| | | val | testA | testB | val | testA | testB | val | test | |
| *Generalist Segmentation Models (≤8B)* | | | | | | | | | | |
| LISA (Lai et al., 2024) | - | 85.4 | 88.8 | 82.6 | 74.2 | 79.5 | 68.4 | 79.3 | 80.4 | 79.8 |
| GSVA (Xia et al., 2024) | - | 86.3 | 89.2 | 83.8 | 72.8 | 78.8 | 68.0 | 81.6 | 81.8 | 80.3 |
| Text4Seg DeepseekVL-1.3B | None | 83.6 | 87.3 | 79.1 | 78.0 | 83.6 | 70.3 | 78.5 | 78.8 | 79.9 |
| Text4Seg DeepseekVL-1.3B | SAM-H | 86.4 | 90.3 | 81.7 | 80.5 | 86.3 | 72.3 | 82.4 | 82.7 | 82.8 |
| Text4Seg DeepseekVL-7B | None | 87.2 | 90.8 | 83.4 | 82.1 | 88.1 | 76.8 | 81.1 | 81.0 | 83.8 |
| Text4Seg DeepseekVL-7B | SAM-H | 89.6 | 93.3 | 85.4 | 84.2 | **90.2** | 78.5 | 84.4 | 84.7 | 86.3 |
| Text4Seg Qwen-VL-7B | None | 87.2 | 90.1 | 83.6 | 82.1 | 87.4 | 76.6 | 81.5 | 81.3 | 83.7 |
| Text4Seg Qwen-VL-7B | SAM-H | 89.7 | 93.0 | 85.8 | 84.6 | 90.1 | 78.6 | 85.0 | 85.1 | 86.5 |
| Text4Seg LLaVA-1.5-7B | None | 89.2 | 92.0 | 86.4 | 83.4 | 88.6 | 78.0 | 81.7 | 82.4 | 85.2 |
| Text4Seg LLaVA-1.5-7B | SAM-H | **90.8** | **93.7** | **87.6** | 84.7 | **90.2** | 79.0 | 84.8 | 85.0 | 87.0 |
| Text4Seg InternVL2-8B | None | 88.3 | 91.4 | 85.8 | 83.5 | 88.2 | 77.9 | 82.4 | 82.5 | 85.0 |
| Text4Seg InternVL2-8B | SAM-H | 90.3 | 93.4 | 87.5 | **85.2** | 89.9 | **79.5** | 85.4 | 85.4 | **87.1** |
| Text4Seg InternVL2-8B[‡] | None | 88.9 | 92.4 | 84.1 | 83.1 | 88.6 | 77.3 | 83.6 | 83.8 | 85.2 |
| Text4Seg InternVL2-8B[‡] | SAM-H | 89.6 | 92.6 | 84.9 | 83.7 | 88.8 | 77.6 | 84.6 | 84.8 | 85.8 |
| *Generalist Segmentation Models (13B)* | | | | | | | | | | |
| Shikra (Chen et al., 2023a) | Vicuna-13B | 87.8 | 91.1 | 81.8 | 82.9 | 87.8 | 74.4 | 82.6 | 83.2 | 84.0 |
| LISA (Lai et al., 2024) | - | 85.9 | 89.1 | 83.2 | 74.9 | 81.1 | 68.9 | 80.1 | 81.5 | 80.6 |
| GSVA (Xia et al., 2024) | - | 87.7 | 90.5 | 84.6 | 76.5 | 81.7 | 70.4 | 83.9 | 84.9 | 82.5 |
| Text4Seg LLaVA-1.5-13B | None | 89.6 | 92.3 | 87.0 | 84.4 | 89.0 | 79.1 | 82.9 | 82.9 | 85.9 |
| Text4Seg LLaVA-1.5-13B | SAM-H | **91.2** | **94.3** | **88.0** | **85.7** | **90.8** | **80.1** | **85.6** | **85.5** | **87.7** |

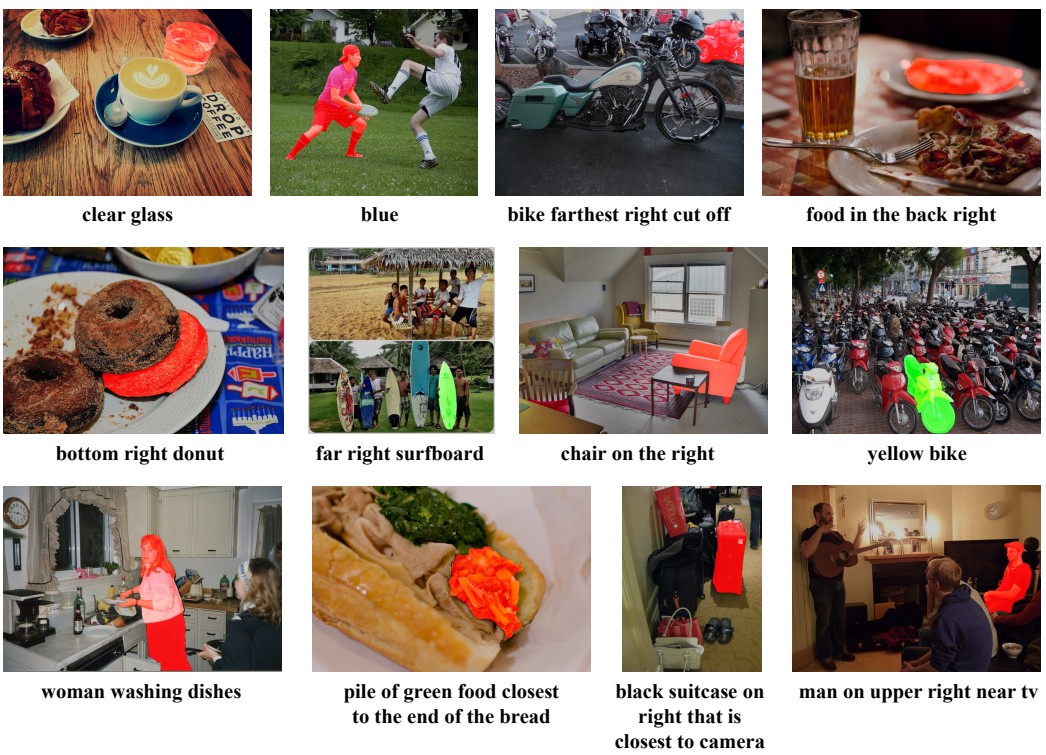

Figure 15: Example results of Text4Seg on referring expression segmentation task. The referring phrases are below the images.

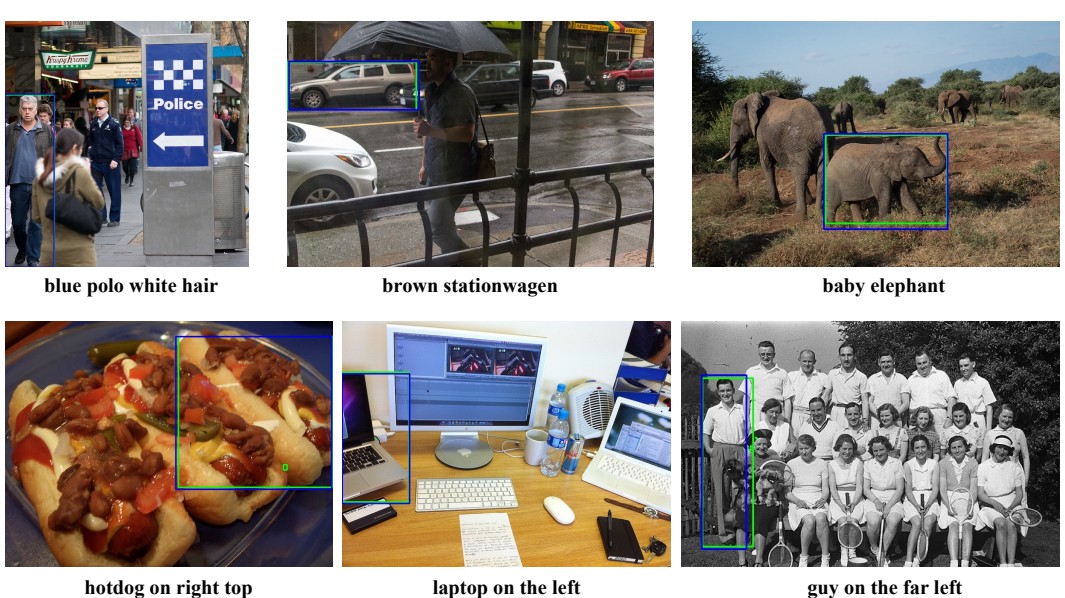

Figure 16: Example results of Text4Seg on referring expression comprehension task. Blue boxes are ground truth labels, and green ones are the Text4Seg predictions.

**Query**: Can you segment the aeroplane, bicycle, bird, boat, bottle, bus, car, cat, chair, cow, dining table, dog, horse, motorbike, person, potted plant, sheep, sofa, train, TV in the image?

**Response**: Sure, the segmented mask is:

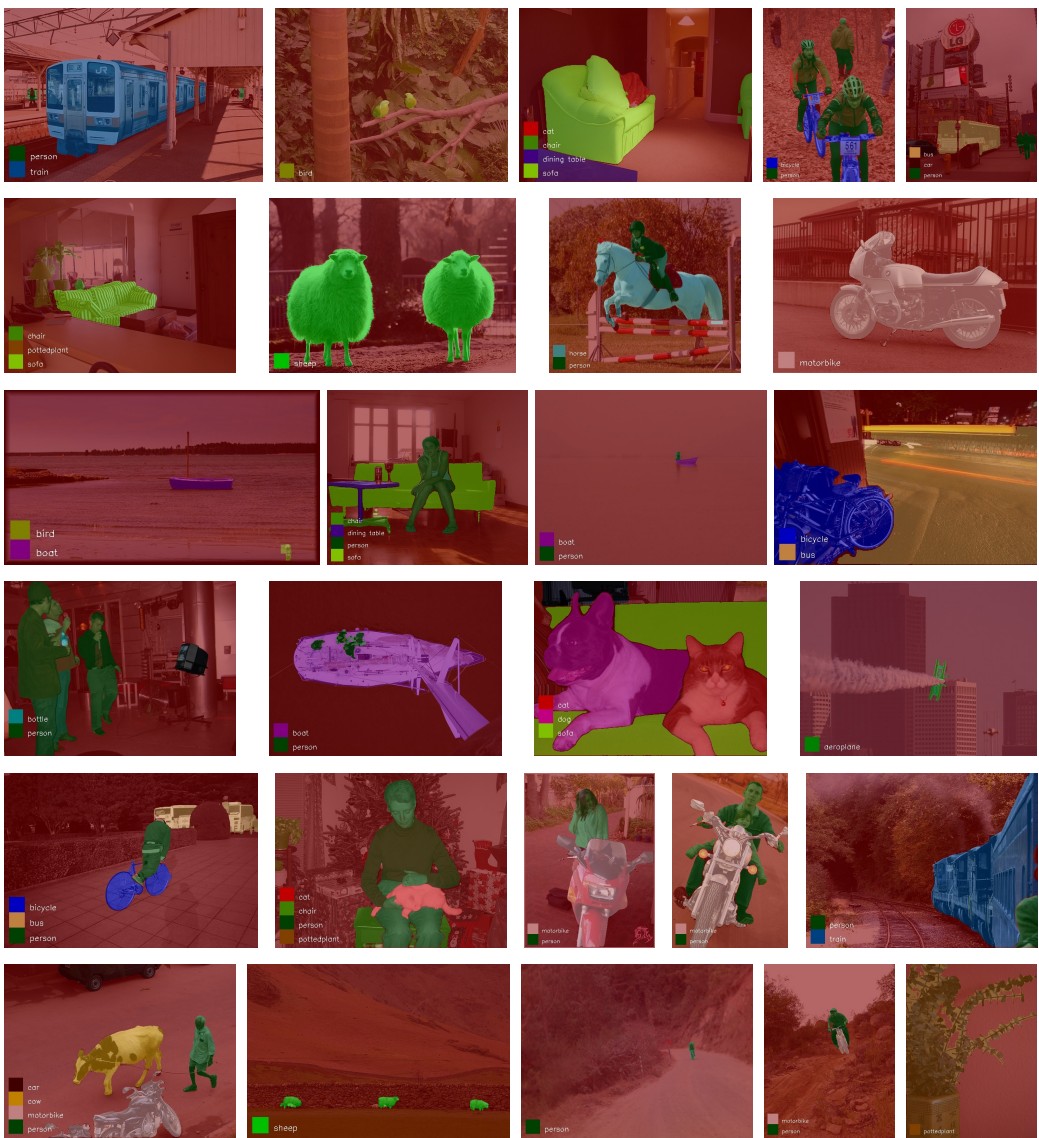

Figure 17: Example results of open-vocabulary segmentation using Text4Seg on the PAS-20 benchmark.

**Query**: Can you segment the <PC-59 CLASS LIST> in the image?

**Response**: Sure, the segmented mask is:

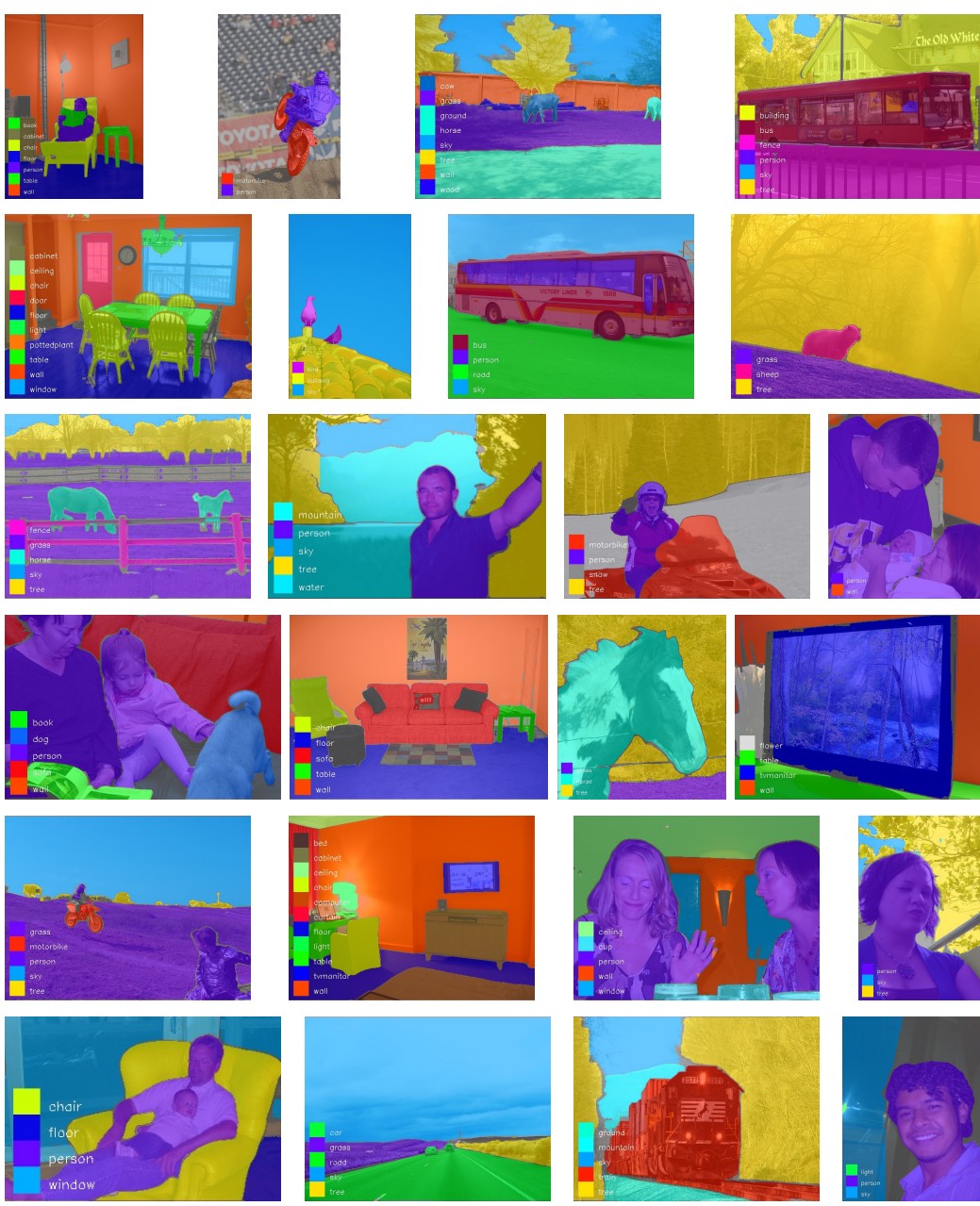

Figure 18: Example results of open-vocabulary segmentation using Text4Seg on the PC-59 benchmark.

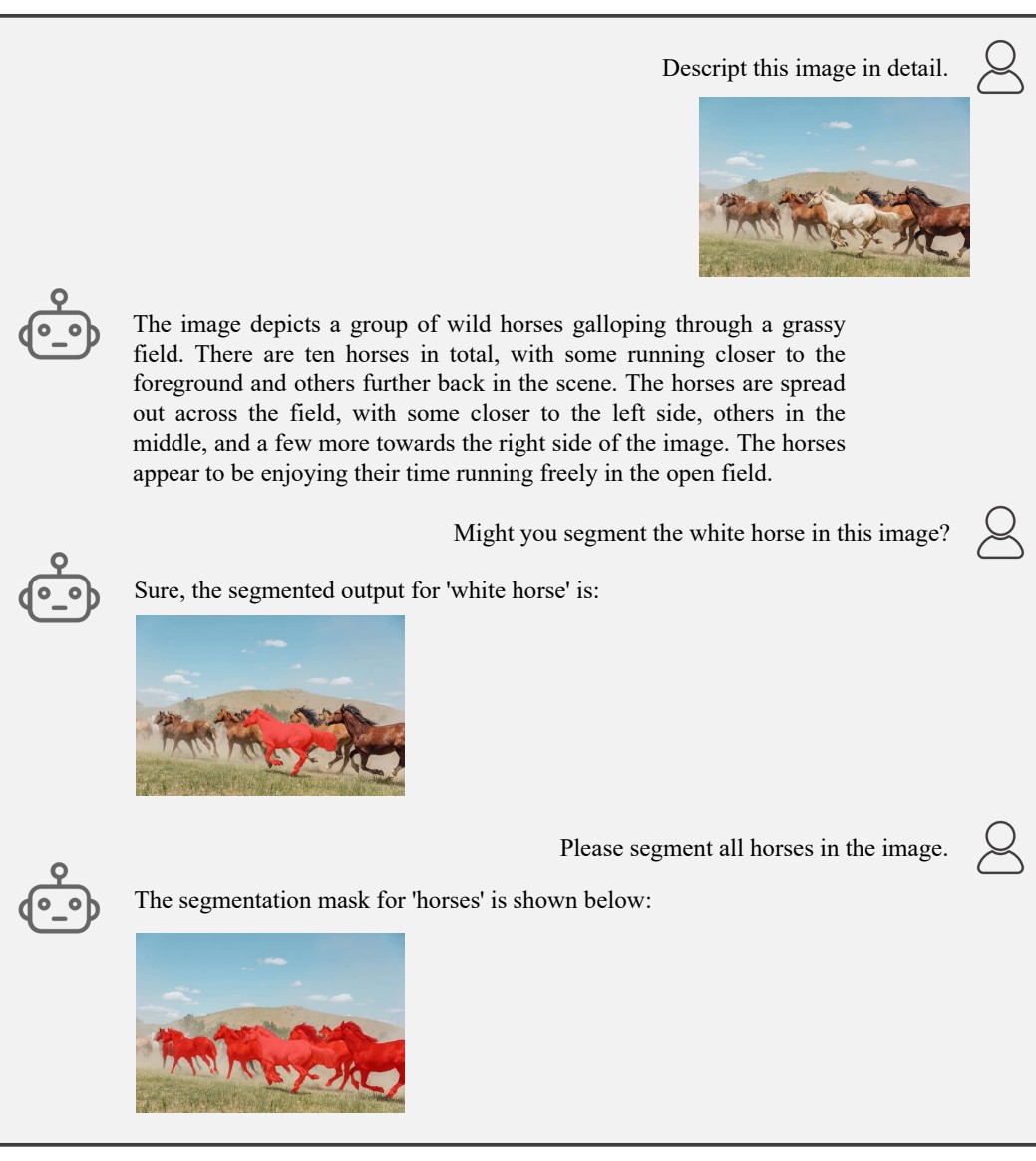

Figure 19: An example result of Text4Seg to perform image captioning, single-object segmentation, and multi-object segmentation.

What is unusual about this image?

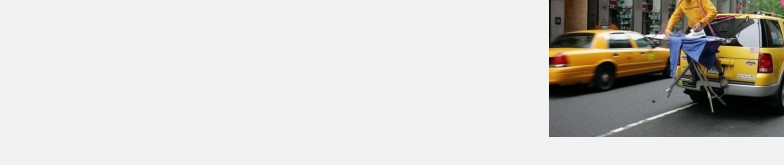

**Text4Seg** The unusual aspect of this image is that a man is standing on the back of a moving yellow car, holding a clothes dryer. This is not a typical scene, as people usually do not stand on the back of a moving vehicle, especially not on top of a car with a clothes dryer. The man's actions are both dangerous and unconventional, as he could risk falling off the car or causing damage to the vehicle or himself.

**LLaVA1.5:** The unusual aspect of this image is that a man is standing on a clothesline mounted on the back of a moving yellow car. This is not a common sight, as people typically do not use cars as a platform for hanging clothes or performing other tasks. The man is also wearing a yellow shirt, which adds to the unconventional nature of the scene. The presence of a taxi in the background further emphasizes the unusual nature of the image, as it is not a typical setting for such an activity

Could you please segment this man? 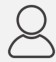

**Text4Seg** Sure, here is the segmentation mask for 'man':

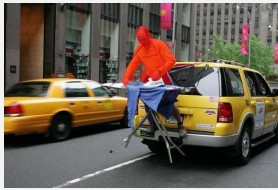

**LLaVA1.5:** 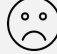

Figure 20: The capability comparison between Text4Seg and LLaVA-1.5.

