# OpenReview forum: "Text4Seg: Reimagining Image Segmentation as Text Generation"
_ICLR.cc/2025/Conference — ICLR 2025 Poster_

### Official Review · Reviewer_stRB · 2024-11-01

**Soundness:** 3
**Presentation:** 3
**Contribution:** 3
**Rating:** 8
**Confidence:** 4

**Summary:**

This paper proposes a new scheme for adapting MLLMs to segmentation by reformulating the mask prediction as a sequence of text labels called *semantic descriptors*. Specifically, the MLLM takes as input an image and query and outputs a mask prediction, represented as a sequence of patch-wise semantic descriptors, consisting of <SEG>, 'others', and the queried target label: <SEG> is a special token that indicates the start and end of the mask, 'others' suggests a background patch, and the target label suggests a target patch. Such formulation allows the MLLM to predict the segmentation task as a sequence of text labels, instead of coordinates or special segmentation tokens, maintaining its original auto-regressive training objective and thereby facilitating easier optimization. Furthermore, the authors propose *Row-wise Run-Length Encoding (R-RLE)* that compresses adjacent repeated tokens in the mask prediction to reduce the number of tokens by 74% and triple the inference speed. The framework, names TextSeg, acheives strong performance on locating tasks like RES, gRES, and REC, showing the location ability of TextSeg, and acheives comparable performance to its MLLM backbone on VQA, suggesting that the MLLM's conversational ability is maintained.

**Strengths:**

Originality
- Authors propose a new formulation of mask predictions by MLLMs. Different from previous work that represent masks as a special <SEG> token or coordinates, this work formulates the mask as a sequence of text labels (called *semantic descriptors*) of 'others' and the queried target label. Such formulation allows the MLLM to be trained for segmentation with the LLM's original autoregressive training objective, allowing easier optimization and the maintanence of the architecture. Although the idea to represent a mask as a sequence of texts itself is not unimaginable, the realization of it is original and novel.
- Authors also propose Row-wise RLE to effectively compress the sequence of semantic descriptors and thereby eliminate any computation burden caused by their new formulation. Although, RLE is an existing technique, the authors effectively adapt it to their configuration, acquiring novelty. Instead of simply proposing a new formulation that would be computationlly heavy by itself, the authors further propose a technique that eradicates this issue, making the formulation applicable and thereby more complete.

Quality
 - Authors carry out an extensive experiment to check whether their new formulation is generalizable to various MLLMs by using four MLLM backbones: LLaVA-1.5 (Li et al., 2024a), Qwen-VL (Bai et al., 2023), DeepseekVL (Lu et al., 2024), and InternVL2 (Chen et al., 2023b).
- Authors also experiment on various tasks (RES, gRES, REC, and Open Vocabulary Segmentation), showing that TextSeg acquires strong performance on various grounding tasks.
- Authors also show that the conversational ability of the LLM does not diminish with the mask prediction finetuning, which is crucial for MLLMs to be useful as its their main capability.

Clarity
 - The paper explains its method and contribution well in overall.

Significance
 - The paper provides a new formulation of mask prediction by LLMs to the research area of adapting MLLMs to grounding tasks, which is of significant interest nowadays, and thus contributes to the community.

**Weaknesses:**

- Although authors stress as a main advantage of their mask formulation as not requiring a segmentation decoder throughout their paper, TextSeg uses SAM to acquire performance comparable to other generalist segmentation models. More precisely, TextSeg *does* require a segmentation decoder but *does not* require finetuning it. Thus, instead of saying that TextSeg is a decoder-free framework, authors should explain that it is a decoder-training-free framework.
- Since the authors say that SAM is optional, the results without SAM should be included in all the main table results, not just in the ablation study.

- Addressing the points below would make the paper clearer:
  - Is the row length in R-RLE 16? Then, is Fig. 3 a smaller version with row length 6? The discrepancy between Fig. 3 and the textual explanation is a bit confusing.
  - Add backbone column to main results table (ie. Table 1~3), so that direct comparison between TextSeg-some-backbone and another method using that backbone is easier to do.
  - Add references of datasets/benchmarks in tables.
  - Explain what 'mix' under 'Training Data' of LISA means in the caption of Table 4.
  - References from the end of page 6 to halfway of page 8 seem to be missing.
  - Explain what training scheme was used for the ablation study: was TextSeg trained on the same combined RES datasets?
  - Typo in "Notably, TextSeg with ViT-L increases the average performance on RES tasks from 70.3 to 75.4 cIoU compared to TextSeg without a mask refiner, with only a little increase in inference time." Should ViT-L be SAM-L? Also, the numbers 70.3 and 75.4 are not in Tab. 6.
  - Performance of 79.3 cIoU of TextSeg with SAM-H in Table 6 is different from performance of 79.2 cIoU in Table 1.
  - Mark best performances in bold in Table 5 (as done in other tables).

**Questions:**

- In Figure 6, higher resolution patch prompts to SAM seems to result in worse segmentation results, and this is counter-intuitive as more finegrained the prompt, more finegrained the segmentation should be. Could the authors provide an explanation?

---

> ### Author Response · Authors · 2024-11-21
> **Author response**
>
> - **W1: Decoder-free framework.**
> As discussed in the general response, our TextSeg does not necessarily rely on the SAM refiner during inference, even though using it may yield better performance. Without SAM decoder, simpler alternatives like Conditional Random Fields (CRF) or $32 \times 32$ semantic descriptors still achieve competitive or superior results compared to existing works. Therefore, we maintain that TextSeg remains a decoder-free framework.
>
> - **W2: Results without SAM.**
> We reported results with SAM in the main table because, by default, we use the off-the-shelf SAM with ViT-H as our mask refiner. Following your suggestion, we attempted to include the results without SAM across all the main tables. However, doing so required excessive space. To address this, we have instead included these results—covering various MLLMs, semantic descriptor lengths, and tasks—in the appendix (Tables 12~14) for comprehensive reference.
>
> - **W3: Improvement points to make paper clearer**
> We greatly appreciate the reviewer’s valuable feedback and have addressed the points as follows:
>     - ***The row length in R-RLE.***
>     By default, the row length in R-RLE is 16, but we used a row length of 6 in Fig. 3 for simplicity. We have clarified this in the section **Definition of semantic descriptors** in our revision.
>     - ***Add backbone column to main results table.***
>     As recommended, we have added an `LMM` column for backbone to the main results tables (Tables 1, 2, and 3).
>     - ***Add references of datasets/benchmarks in tables.***
>     We have included references for each dataset in the captions of Tables 1, 2, and 3 as suggested.
>     - ***Explain what 'mix' under 'Training Data' of LISA means.***
>     The term "Mix" under LISA’s training data denotes a blend of different datasets proposed in LISA, including semantic segmentation datasets, referring segmentation datasets, and visual question answering datasets. For more details, please see the "Training Datasets of Continued Pre-Training (CPT)" table or refer to the LISA paper. This clarification has been added to the caption of Table 4 in the revised manuscript.
>     - ***References from the end of page 6 to halfway of page 8 seem to be missing***
>     Thanks for your suggestion. The omitted references in the main text are already cited in Table 3. To maintain readability, we have avoided duplicating these references.
>     - ***Training scheme was used for the ablation study.***
>     For the ablation studies on **resolution of semantic descriptors** and **mask refiner with SAM variants**, TextSeg was trained using the same combined RES datasets as in the main experiments. For the ablation study of **I-RLE v.s. R-RLE**, TextSeg was trained on the train split of the refCOCO and refCOCO+ datasets. The refCOCOg dataset was excluded in this case due to its longer referring sentences, which lead to excessively long semantic descriptors when compression techniques are not applied.
>     - ***Typos and incorrect numbers.***
>     Thank you for pointing out these careless mistakes. We have corrected them in our revision as below:
>         > Notably, TextSeg with SAM-L improves the average performance on RES tasks from 73.5 to 79.1 cIoU compared to TextSeg without a mask refiner, with only a minimal increase in inference time.
>
>         We have also updated the performance in Table 6 from 79.3 to 79.2 cIoU.
>     - ***Mark best performances in bold in Table 5.***
>     We have bolded the best performances in Table 5 in the latest revision.
>
> - **Q1: Explanation about worse segmentation results of higher resolution.**
> There are two primary factors that can affect the segmentation performance.
>    - The first is the **visual grounding ability**, which refers to the model's capacity to follow instructions and to localize the position of objects.
>    - The second factor is the **shape boundary accuracy** of the predicted mask with the groundtruth mask.
>
>   While TextSeg, with higher-resolution semantic descriptors, can produce more fine-grained masks, increasing descriptor length also poses challenges for model training. This may negatively impact the model's visual grounding ability, particularly when compared to a model using a $16\times16$ resolution.
>
>   Since the SAM refiner mainly helps to increase the shape boundary accuracy, it may not be that effective when the visual grounding performance is worse. Therefore, the results of $32\times32$ may be sometimes worse than those of $16\times16$ due to the decrease of visual grounding ability. We anticipate that this limitation could be addressed through further scaling of the model and dataset, enabling improvements in both grounding ability and boundary accuracy.

---

> > ### Comment · Reviewer_stRB · 2024-11-24
> >
> > Thank you for your explanation. Most of my concerns are resolved.
> >
> > However, regarding W1 and W2, I still believe that the paper should either present TextSeg as 'decoder-training-free' or include results w/o SAM in the main table.
> > Authors present TextSeg as 'decoder-free' throughout the paper, but, as the authors explain in the rebuttal to W2, their **default** is TextSeg **with** an "off-the-shelf SAM with ViT-H as our mask refiner". Explaining that the framework is decoder-free and then including results with a decoder in the main table is inconsistent and thus can be misleading to readers. My concern is not about fairness or whether this framework works well without SAM (as shown, TextSeg still attains competitive performance with postprocessing methods or higher resolution descriptors instead of SAM) but the presentation. Thus, I suggest the authors to describe their framework as 'decoder-training-free' if they are to include only the TextSeg with SAM results in the main table or include results w/o SAM in the main table.

---

> ### Author Response · Authors · 2024-11-25
>
> Thank you for your valuable and constructive feedback that really help improve the quality of our paper. We agree that including results without SAM helps to make our paper clearer. We therefore have implemented the following revisions:
>
> 1. **Inclusion of Results Without SAM in Main Tables**
> To address your suggestion, we have included results without the SAM (using CRF or higher resolution of semantic descriptors) for RES and GRES in Tables 1&2, results without any mask refiner for REC in Table 3. Kindly refer to the latest revision for details.
> 2. **Additional Discussion in Section "Result of Single Object" (Table 1, RES)**
> We have added further discussion to highlight the competitiveness of our method without SAM refiner:
>    > Even without using the SAM refiner, our method remains competitive. For instance,  TextSeg_{\textup{InternVL2-8B}}^\dagger, refined with CRF, and TextSeg$_{\textup{InternVL2-8B}}$$^\ddagger$, based on $32\times32$ semantic descriptors, achieve results that rival or exceed existing methods.
> 3. **Additional Discussion in Section "REC: Results" (Table 3, REC)**
> We have also included additional insights regarding REC results:
>    > It is worth noting that TextSeg_{\textup{LLaVA-1.5-7B}}* and TextSeg$_{\textup{LLaVA-1.5-13B}}$$^*$, without a mask refiner, outperform their respective baseline counterparts. These results emphasize the superiority of TextSeg in following instructions, leading to enhanced visual grounding ability.
>
> We hope these revisions address your concerns and provide greater clarity to our analysis.

---

> > ### Comment · Reviewer_stRB · 2024-11-26
> >
> > I have read the revised manuscript and my concerns are resolved. I will raise my score.
> > (Also, the dagger size seem to vary within Table 2 which the authors may want to check out).

---

> > > ### Author Response · Authors · 2024-11-26
> > >
> > > We sincerely appreciate your valuable feedback and the recognition of our work.

---

### Official Review · Reviewer_SUs4 · 2024-11-03

**Soundness:** 2
**Presentation:** 3
**Contribution:** 2
**Rating:** 5
**Confidence:** 2

**Summary:**

This paper proposed TextSeg, a novel text-as-mask paradigm that casts image segmentation as a text generation problem, eliminating the need for additional decoders. It employs semantic descriptors as a new textual representation of segmentation masks where each image patch is mapped to its corresponding text label. It further compresses predicted token length by using Row-wise Run-Length Encoding to handle contiguous region of same semantic region.

**Strengths:**

- model pixel label as semantic text token by generalized VLLM model
- compress token length with row-wise run-length encoding
- comprehensive results on referring express segmentation and comprehension

**Weaknesses:**

- idea is simple and straightforward
- relies on SAM to get the final pixel level prediction from patch level semantic text token prediction
- should compare with other VLLM approach also with SAM refinement

**Questions:**

- please discuss how to extend to instance level segmentation
- the name of TextSeg was already used: https://paperswithcode.com/dataset/textseg

---

> ### Author Response · Authors · 2024-11-21
> **Author response**
>
> - **Rely on SAM to get the final pixel level prediction.**
> Please refer to the general response.
>
> - **Compare with other VLLM approach also with SAM refinement.**
> Please refer to the general response.
>
> - **Extend to instance level segmentation.**
> Referring expression segmentation models inherently perform segmentation at the instance level, as they are capable of isolating a specific instance from multiple objects. For example, as shown in Figure 9 of our paper, a prompt like `tallest young giraffe` identifies and segments the tallest giraffe among four giraffes. Similarly, a prompt like `green boy not kicking` focuses on the individual in a green t-shirt who is not kicking, out of six people in the image.
> Furthermore, our method can be easily adapted for traditional instance segmentation tasks. For example, a user query might be: `Could you segment all the people in the image and label each individual separately?` The corresponding response could be structured as: `<seg> others, ..., person 1, ..., person 2, ..., person 3, ..., others </seg>`.
> In this manner, our method can be extended to traditional instance segmentation. Semantic descriptors can be constructed directly from existing instance segmentation datasets, making our approach versatile and easily adaptable to various instance-level segmentation tasks.
>
> - **The name of TextSeg.**
> We have revised the name of our method from TextSeg to Text4Seg in our latest revision. However, for the purpose of this rebuttal, we continue to use the name TextSeg to maintain clarity and avoid confusion for the reviewers.

---

> ### Author Response · Authors · 2024-12-01
>
> Dear Reviewer SUs4,
>
> Thank you for your valuable feedback on our submission. We have addressed the concerns raised in your initial review, and we would like to confirm if there are any additional questions or points that require further clarification from our side.
>
> If our responses have satisfactorily addressed your concerns, we kindly request your reconsideration of the overall assessment. We greatly appreciate your time and effort in reviewing our work and your contributions to improving its quality.

---

### Official Review · Reviewer_9zHz · 2024-11-03

**Soundness:** 2
**Presentation:** 3
**Contribution:** 2
**Rating:** 6
**Confidence:** 4

**Summary:**

The paper introduces TextSeg, a novel approach that integrates image segmentation into Multimodal Large Language Models (MLLMs) by framing segmentation as a text generation task.  The key innovation is the use of semantic descriptors, where each image patch is mapped to a textual label, allowing seamless integration into the auto-regressive training pipeline of MLLMs. By representing images with 16×16 semantic descriptors, the approach achieves competitive segmentation performance. To enhance efficiency, the authors propose Row-wise Run-Length Encoding (R-RLE), which compresses redundant text sequences by 74% and speeds up inference by three times without sacrificing accuracy. Extensive experiments demonstrate that TextSeg attains state-of-the-art results on multiple datasets across various vision tasks.

**Strengths:**

The authors have conducted extensive experiments and ablation studies to demonstrate the effectiveness of their proposed method.

**Weaknesses:**

The paper presents a method that integrates image segmentation into MLLMs by introducing semantic descriptors and utilizing a SAM mask refiner. While the approach simplifies the segmentation process by treating it as a text generation task, the technical contributions appear to be more incremental and engineering-oriented. The method essentially adapts existing MLLMs with semantic descriptors to perform segmentation tasks, serving as a baseline framework that can be applied to different MLLM models.

**Questions:**

+ Training Datasets: The authors should clearly mention about the training datasets, additional training datasets for fine-tuning (if accapable) of other methods for a fair comparison. Because we know that with more training datasets (GLaMM, Groundhog), and with fine-tuning on task-specific datasets, we can achieve a better performance.
+ Use of SAM Mask Refiner: The proposed TextSeg method incorporates a post-processing step using the SAM mask refiner. As indicated in Table 6, without SAM, the method achieves a cIoU of 73.5, which is lower than most methods in the Generalist Segmentation Models (~7B) category in Table 1. This raises concerns about the fairness of the comparison, as other methods may not use similar post-processing techniques. The authors should clarify how other methods could benefit from similar enhancements.
+ Inference Speed: It would be beneficial to include the inference speed of TextSeg compared to other methods. Providing quantitative metrics on inference time would strengthen the evaluation.
+ Missing Recent Specialized Segmentation Models: The paper lacks comparisons with recent specialized segmentation models such as PolyFormer [1], UNINEXT [2], and HIPIE [3]. Including these models in the evaluation would provide a more comprehensive assessment of the proposed method's performance relative to the current state-of-the-art.

[1] Liu et al., PolyFormer: Referring image segmentation as sequential polygon generation, CVPR, 2023.

[2] Yan et al., Universal instance perception as object discovery and retrieval, CVPR, 2023.

[3] Wang et al., Hierarchical open-vocabulary universal image segmentation, NeurIPS, 2024.

---

> ### Author Response · Authors · 2024-11-21
> **Author response (1/2)**
>
> **Technical Contributions**
>
> Our main contribution lies in introducing the _text-as-mask_ paradigm, driven by novel semantic descriptors, which redefines image segmentation as a text generation problem.  The objective of our work is to equip existing MLLMs with segmentation capabilities without necessitating architectural modifications. As such, **we intentionally avoid making technical modification to existing MLLMs**, as our focus is on harnessing their inherent text generation capabilities to achieve segmentation in a streamlined and efficient manner.
>
> **Q1: Training Datasets**
>
> Most prior methods follow a two-stage training paradigm: **Continued Pre-Training (CPT)** using large datasets, followed by **Supervised Fine-Tuning (SFT)** for specific tasks. The datasets used in these approaches are summarized in the following tables:
>
> - **Table R1-1:** Datasets for **Continued Pre-Training (CPT)**
> - **Table R1-2:** Datasets for **Supervised Fine-Tuning (SFT)** in **Referring Expression Segmentation (RES)**
> - **Table R1-3:** Datasets for **Supervised Fine-Tuning (SFT)** in **Generalized Referring Expression Segmentation (GRES)**
>
> We can note that:
> 1. For CPT, previous methods rely heavily on large and diverse datasets, whereas our approach, **TextSeg**, eliminates this requirement, demonstrating superior efficiency and effectiveness.
> 2. For SFT, we ensure a fair comparison by following previous works and trained on:
>    - The `train` split of refCOCO series for **RES** and **REC** tasks.
>    - The `train` split of grefCOCO for the **GRES** task.
>
> _We have also included this discussion in the Appendix, under the section **Comparison of training datasets**, in our latest revision._
>
> Table R1-1: Training datasets of **Continued Pre-Training (CPT)**.
> | Methods | Datasets |
> | -------- | -------- |
> | LISA      |  ADE20K, COCO-Stuff,  PACO-LVIS, PartImageNet, PASCAL-Part, refCLEF,  refCOCO, refCOCO+, refCOCOg, LLaVA-v1.5-mix665k|
> | PixelLM    | ADE20K, COCO-Stuff,  PACO-LVIS, refCLEF,  refCOCO, refCOCO+, refCOCOg, LLAVA-150k, multi-target reasoning segmentation (MUSE)       |
> | GSVA      | ADE20K, COCO-Stuff, PACO-LVIS, Maplilary Vistas, PASCAL-Part, refCLEF,  refCOCO, refCOCO+, refCOCOg, gRefCOCO, LLaVA-Instruct-150K, ReasonSeg|
> | AnyRef    | ADE20K, COCO-Stuff,  PACO-LVIS, refCLEF,  refCOCO, refCOCO+, refCOCOg, PhraseCut, Flickr30K Entities, AVSBench |
> | NEXT-Chat | Flickr30K Entities, Visual Genome, RefCOCO, RefCOCO+, RefCOCOg, VQAv2, PointQA, Visual7W, VCR, LLaVA-Instruct-150K, VG grounded captioning, Shikra-RD |
> | Groundhog | Multi-Modal Multi-Grained Grounding dataset (M3G2): PNG, Flickr30K-Entity, refCLEF,  refCOCO, refCOCO+, refCOCOg, gRefCOCO, PhraseCut, D-Cube, ReasonSeg, RIO, SK-VG, VizWiz-G, TextVQA-X, GQA, VQS, Shikra-BinaryQA, EntityCount, FoodSeg-QA, LVIS-QA, RefCOCO-REG, RefCOCO+-REG, RefCOCOg-REG, gRefCOCO-REG, VG-SpotCap, V7W, PointQA, VCR, ShikraRD, SVIT-RD, Guesswhat, VG-RefMatch, HierText |
> | GLaMM     | Grounding-anything Dataset (GranD): **11M** images, **810M** masks, **84M** referring expressions, GranD-f |
> | TextSeg   | **Our method does not need CPT** |
>
> Table R1-2: Referring Expression Segmentation Datasets of **Supervised Fine Tuning (SFT)**. $^*$Other methods have already incorporated refCLEF dataset in their CPT training datasets.
> | Methods | Datasets |
> | -------- | -------- |
> | LISA      | refCOCO, refCOCO+, refCOCOg |
> | PixelL    | None     |
> | GSVA      | refCOCO, refCOCO+, refCOCOg |
> | AnyRef    | refCOCO, refCOCO+, refCOCOg |
> | NEXT-Chat | refCOCO, refCOCO+, refCOCOg |
> | Groundhog | None     |
> | GLaMM     | refCOCO, refCOCO+, refCOCOg     |
> | TextSeg   | refCOCO, refCOCO+, refCOCOg, refCLEF$^*$  |
>
> Table R1-3: Generalized  Referring Expression Segmentation Datasets of **Supervised Fine Tuning (SFT)**.
>
> | Methods | Datasets   |
> | -------- | --------  |
> | LISA      | grefCOCO |
> | GSVA      | grefCOCO |
> | TextSeg   | grefCOCO |
>
> **Q2: Use of SAM Mask Refiner**
>
> Please refer to the general response.

---

> ### Author Response · Authors · 2024-11-21
> **Author response (2/2)**
>
> **Q3: Inference Speed**
>
> We report the comparison of inference time (second per sample) as well as the image size in Table R1-4. TextSeg achieves an inference time of 5.92 seconds per sample, which is competitive compared to PixelLM and NEXT-Chat. Although slower than GSVA (0.42 seconds), this is due to GSVA processing smaller image sizes ($224^2$).
>
> Table R1-4: Inference time comparison.
> | Methods | LISA | GSVA |  PixelLM | NEXT-Chat | TextSeg (InternVL2) |
> | -------- | --------  | -------- | --------  | -------- | -------- |
> | Image size | $336^2$ | $224^2$ | $448^2$ | $336^2$ | $336^2$
> |  Time (s) | 3.82 | 0.42 | 6.65 | 6.20 | 5.92 |
>
> **Q4: Missing Recent Specialized Segmentation Models**
>
> Thank you for your suggestion. We have included these models in Table 1 and Table 3 in our latest revision to ensure a more comprehensive comparison, as well as in Table R1-5 and Table R1-6 below. Notably, our TextSeg still demonstrates competitive performance compared to these specialized segmentation models.
>
> Table R1-5: Referring Expression Segmentation results (cIoU) on RefCOCO (+/g) datasets.
>
> |  | val | testA | testB | val(+) | testA(+) | testB(+) | val(g) | test(g) | Avg. |
> | -------- | -------- | -------- | -------- | -------- | -------- | -------- | -------- | -------- | ------- |
> | HIPIE | 78.3 | - | -| 66.2 | - | - | 69.8 | - | - |
> | PolyFormer-L | 76.0 | 78.3 | 73.3 | 69.3 | 74.6 | 61.9 | 69.2 | 70.2 | 71.6 |
> | UNINEXT-L | 80.3 | 82.6 | 77.8 | 70.0 | 74.9 | 62.6 | 73.4 | 73.7 | 74.4 |
> | TextSeg (InternVL2) | 79.2 | 81.7 | 75.6 | 72.8 | 77.9 | 66.5 | 74.0 | 75.3 | 75.4 |
>
> Table R1-6: Referring Expression Comprehension results (Acc@0.5) on RefCOCO (+/g) datasets.
>
> |  | val | testA | testB | val(+) | testA(+) | testB(+) | val(g) | test(g) | Avg. |
> | -------- | -------- | -------- | -------- | -------- | -------- | -------- | -------- | -------- | ------- |
> | PolyFormer-L | 90.4 | 92.9 | 87.2 | 85.0 | 89.8 | 78.0 | 85.8 | 85.9 | 86.9 |
> | UNINEXT-L | 91.4 | 93.7 | 88.9 | 83.1 | 87.9 | 76.2 | 86.9 | 87.5 | 87.0 |
> | TextSeg (InternVL2) | 90.3 | 93.4 | 87.5 | 85.2 | 89.9 | 79.5 | 85.4 | 85.4 | 87.1 |

---

> > ### Comment · Reviewer_9zHz · 2024-11-26
> >
> > I would like to thank the authors for addressing my concerns. Below is my response:
> > - Regarding to the technical contribution, it still appears to be more incremental and engineering-oriented.
> > - Q1: Thank you for providing datasets' information.
> > - Q2: We can clearly see that without any mask refiner (including SAM or CRF), the performance dropped significantly and just on the par or in some cases, lower than previous methods.
> > - Q3: if only considering the methods with the same image size, the proposed method is a bit better than NEXT-Chat, but notably slower than LISA, and I believe that the same thing will happen to PixelLM if we consider the same image size.
> > - Q4: If we consider full version of the proposed method, they outperform the new suggested methods and the mentioned methods in the initial manuscript. However, if we consider the non mask refiner variant of the proposed method (for a fair comparison as other methods don't have post-processing steps), the proposed method is just on par or lower than previous methods.

---

> > > ### Author Response · Authors · 2024-12-01
> > >
> > > Dear Reviewer 9zHz,
> > >
> > > Thank you very much for your valuable feedback and constructive comments on our submission. We greatly appreciate the time and effort you have devoted to reviewing our work.
> > >
> > > With regard to the technical contribution, we acknowledge that our focus is less on proposing architectural innovations and more on introducing the _text-as-mask_ paradigm. Our primary objective is to showcase how existing MLLMs can be effectively adapted for segmentation tasks without requiring architectural modifications, emphasizing conceptual clarity and practical applicability.
> > >
> > > We have carefully revised our manuscript in response to your suggestions and addressed the concerns raised in your review. If there are any additional questions or issues that you feel require clarification or further improvement, we would be more than happy to address them.
> > >
> > > Thank you again for your thoughtful review and for helping us enhance the quality of our work.

---

> ### Author Response · Authors · 2024-11-26
>
> Thank you for your detailed feedback and for sharing your concerns. We deeply appreciate the opportunity to address them in further detail below.
>
> > Q2: We can clearly see that without any mask refiner (including SAM or CRF), the performance dropped significantly and just on the par or in some cases, lower than previous methods.
>
> **Without any mask refiner**, as a lower bound, our TextSeg ($32^2$) achieves 71.4 cIoU (Table 1) on average on the RES tasks, **outperforming most existing methods**, including NEXT-Chat (68.9), LISA (69.9), PixelLM (69.2), LaSagnA (71.1), and equaling GSVA (71.4). Additionally, it achieves an average score of 67.6 (Table 2) on the GRES tasks, **higher than all comparisons**, including LISA (62.9) and GSVA (65.6). Lastly, on REC tasks, it achieves an average score of 85.0 (Table 3) , **better than most comparisons**, including Shikra (82.9), Ferret (83.9), LISA (79.8), GSVA (80.3), NEXT-Chat (80.4), which indicates a better visual grounding ability of TextSeg and further explains its superiority in RES and GRES (require strong visual grounding and shape boundary accuracy). Not to mention, it is entirely fair to compare our TextSeg with a mask refiner to enhance shape boundary accuracy against these existing methods.
>
> > Q3: if only considering the methods with the same image size, the proposed method is a bit better than NEXT-Chat, but notably slower than LISA, and I believe that the same thing will happen to PixelLM if we consider the same image size.
>
> **We have not claimed in our manuscript that TextSeg offers faster inference than other methods**. In fact, it achieves comparable efficiency to several existing approaches. Additionally, PixelLM’s design mandates an image size of 448; reducing the size will degrades its performance.
>
> > Q4: If we consider full version of the proposed method, they outperform the new suggested methods and the mentioned methods in the initial manuscript. However, if we consider the non mask refiner variant of the proposed method (for a fair comparison as other methods don't have post-processing steps), the proposed method is just on par or lower than previous methods.
>
> **All** the specialized segmentation models and generalized segmentation models **in comparison** inherently **include a decoder** in their architectures, which will be optimized during training. This is why **additional post-processing is neither necessary nor effective for them**. We question the rationale for restricting our method from utilizing a simple off-the-shelf mask refiner, merely because previous methods do not require it. If we were to exclude the SAM decoder and other post-processing steps from our method, then for a fair comparison, should the decoders in all other methods also be removed during evaluation?
>
>
> `We find it difficult to understand why it is necessary for the lower bound (non-mask refiner variant) of our method to outperform the optimal performance (with decoder) of all other methods.` We believe this was not claimed in our manuscript.

---

### Author Response · Authors · 2024-11-21
**General response (2/2)**

Table 3: Generalized Referring Expression Segmentation results on the gRefCOCO dataset.


|  | Refiner | Val (gIoU)  | Val (cIoU)  | Test A (gIoU) | Test A (cIoU) | Test B (gIoU) | Test B (cIoU)  | Avg. |
| -------- | -------- | -------- | -------- | -------- | -------- | -------- | -------- | -------- |
| GSVA | w/o | 66.5 | 63.3 | 71.1 | 69.9 | 62.2 | 60.5 | 65.6 |
| GSVA | SAM | 64.5 | 57.9 | 68.5 | 64.9 | 60.7 | 55.7 | 62.0 |
| TextSeg ($16^2$) | w/o | 68.8 | 63.1 | 66.9 | 67.1 | 62.1 | 61.6 | 64.9 |
| TextSeg ($16^2$) | SAM | 74.4 | 69.1 | 75.1 | 73.8 | 67.3 | 66.6 | 71.1 |
| TextSeg ($16^2$) | CRF | 70.0 | 66.1 | 69.4 | 70.9 | 63.1 | 64.1 | 67.3 |
| TextSeg ($32^2$) | w/o | 71.8 | 65.6 | 71.2 | 70.0 | 64.2 | 62.5 | 67.6 |

### Appendix CRF code

```python
import pydensecrf.densecrf as dcrf
from pydensecrf.utils import unary_from_labels

def crf_refine(image, mask, n_mask=2, gt_prob=0.9):
    """
    Refine a coarse mask using DenseCRF.

    :param image: Input image (HxWx3)
    :param mask: Coarse mask (HxW)
    :return: Refined mask
    """
    # Create the DenseCRF model
    h, w = image.shape[:2]
    mask = mask.flatten()
    d = dcrf.DenseCRF2D(w, h, n_mask)

    # Get unary potentials (negative log probabilities)
    unary = unary_from_labels(mask, n_mask, gt_prob=gt_prob, zero_unsure=False)
    d.setUnaryEnergy(unary)

    # This adds the color-independent term, features are the locations only.
    d.addPairwiseGaussian(sxy=(3, 3), compat=3, kernel=dcrf.DIAG_KERNEL,
                          normalization=dcrf.NORMALIZE_SYMMETRIC)

    # This adds the color-dependent term, i.e. features are (x,y,r,g,b).
    d.addPairwiseBilateral(sxy=(50, 50), srgb=(13, 13, 13), rgbim=img,
                           compat=10,
                           kernel=dcrf.DIAG_KERNEL,
                           normalization=dcrf.NORMALIZE_SYMMETRIC)

    # Inference
    refined_mask = d.inference(5)  # Number of iterations
    refined_mask = np.argmax(refined_mask, axis=0)

    return refined_mask
```

---

### Author Response · Authors · 2024-11-21
**General response (1/2)**

We sincerely thank all reviewers’ and ACs’ for their time, effort, and thoughtful feedback on our paper. We deeply appreciate the reviewers' recognition of the effectiveness and originality of our contributions. **We have provided the latest revision in the system, with the changes highlighted in blue for clarity.** Below, we address the common concern raised by all reviewers:

**Use of SAM as a Mask Refiner**  (@9zHz, SUs4, stRB)

We understand the concerns regarding fairness in using SAM. However, we want to emphasize that our **TextSeg** does not rely on SAM to achieve better performance than existing works. Instead, we employ SAM as an **offline postprocessing** method primarily because it is more effective than other techniques such as CRF. Below, we provide a more detailed explanation:

1. **Fairness of comparing with existing methods**
In Table 1, we summarise the segmentation decoders used in existing _embeddings-as-mask_ methods and observe that the comparison methods already incorporate SAM (or similar pixel decoders) in their architectures. From this standpoint, **it is entirely fair and justified  to directly compare our TextSeg with a SAM refiner against these existing methods.**

2. **Performance of existing methods with SAM postprocessing**
We also try to apply the same post-processing technique to the comparison methods. We chose GSVA as a representative due to its available checkponts that could completely reproduce the results on RES and GRES tasks. As shown in Tables 2 and 3, incorporating an additional SAM refiner into GSVA did not yield improved results. Despite experimenting with different combinations of prompts for SAM refiner in GSVA, we were unable to achieve better performance. We hypothesize that the reason may be that GSVA already optimizes the SAM decoder during training to achieve optimal performance, and adding an additional SAM refiner may, in turn, negatively affect its overall performance. Therefore, **there is no clear evidence that adding a SAM refiner would enhance the performance of existing _embeddings-as-mask_ methods**, as these methods already optimize their SAM decoders during training.

3. **Outstanding performance of TextSeg without SAM**
We provides two solutions to improve the performance of TextSeg (InternVL2-8B) without the SAM refiner.
   - We could use the traditional mask refinement techniques such as Conditional Random Fields (CRF) to produce more fine-grained segmentation masks. As shown in Tables 2 and 3, applying CRF improves TextSeg without SAM from an average cIoU of 67.5 to 70.1 on the RES task, compared to GSVA's 71.4 which uses SAM as decoder in its architecture. Similarly, for the GRES task, **the average score improves from 64.9 to 67.3, surpassing GSVA's 65.6**.
    - Another solution is to use $32 \times 32$ semantic descriptors without any postprocessing. This makes TextSeg directly generate more fine-grained masks from MLLM. **It boosts average performance to 71.4 cIoU on the RES tasks, outperforming most existing methods, including NEXT-Chat (68.9), LISA (69.9), PixelLM (69.2), LaSagnA (71.1), and equaling GSVA (71.4).** On the GRES tasks, TextSeg with $32 \times 32$ semantic descriptors achieves **an average score of 67.6, higher than LISA's 62.9 and GSVA's 65.6**.
**These results demonstrate the simplicity and effectiveness of our TextSeg approach, even without relying on a SAM refiner.**


Table 1: Summary of decoders of existing methods.
|   | LISA | GSVA | PixelLM | NEXT-Chat | AnyRef | LaSagnA | Groundhog | GLaMM |
| -------- | -------- | -------- | -------- | -------- | -------- | -------- | -------- | -------- |
| Decoder | SAM  | SAM  | Customized decoder  | SAM  | SAM  | SAM  | Mask2Former+  | SAM  |

Table 2: Referring Expression Segmentation results (cIoU) on RefCOCO (+/g) datasets.
|  | Refiner | val | testA | testB | val(+) | testA(+) | testB(+) | val(g) | test(g) | Avg. |
| -------- | -------- | -------- | -------- | -------- | -------- | -------- | -------- | -------- | ------- | ------- |
| GSVA | w/o | 77.2 | 78.9 | 73.5 | 65.9 | 69.6 | 59.8 | 72.7 | 73.3 | 71.4 |
| GSVA | SAM | 70.8 | 73.4 | 66.7 | 60.7 | 64.9 | 55.2 | 67.3 | 67.9 | 65.9 |
| TextSeg ($16^2$) | w/o | 70.3 | 71.9 | 68.7 | 65.0 | 68.9 | 60.8 | 66.7 | 67.6 | 67.5 |
| TextSeg ($16^2$) | SAM | 79.2 | 81.7 | 75.6 | 72.8 | 77.9 | 66.5 | 74.0 | 75.3 | 75.4 |
| TextSeg ($16^2$) | CRF | 73.0 | 75.2 | 70.7 | 67.6 | 72.1 | 62.6| 68.9 |70.3 | 70.1 |
| TextSeg ($32^2$) | w/o | 74.7 | 77.4 | 71.6 | 68.5 | 73.6 | 62.9 | 70.7 | 71.6 | 71.4 |

---

### Meta-Review · Area_Chair_Hqkd · 2024-12-17

**Metareview:**

The reviewers appreciated the novel and sound approach to segmentation using MLLM (stRB), the reasonable design of the proposed model (SUs4, stRB), extensive experiments (9zHz, SUs4, stRB), and clarity (stRB) of te paper. They however unanimously raised concerns about the use of SAM mask refiner that could cause fairness issues in comparisons with prior arts and contradict one of the main advantages the authors claimed in the paper. They also pointed out limited technical contribution (9zHz, SUs4), lack of comparisons with relevant methods (e.g., recent specialist segmentation models and other MLLMs) (9zHz, SUs4), and some presentation issues (stRB).

The authors' rebuttal, revision, and subsequent responses in the discussion period addressed most of these concerns. Consequently, two of the three reviewers were supportive of the paper after the discussion period, and while the other reviewer's rating remains a borderline rejection, that reviewer noted that his/her major concern with the use of SAM mask refiner has been successfully resolved.

The AC found that the strengths of the paper, particularly the novel approach and extensive experiments, and the rebuttal outweigh the remaining concerns, and thus recommend acceptance of the paper. The authors are strongly encouraged to carefully revise the paper to reflect the valuable comments by the reviewers, to add new results brought up in the rebuttal and discussions, and to further improve the quality of writing.

**Additional Comments On Reviewer Discussion:**

Two reviewers supported the paper after active discussions with the authors. The other reviewer, i.e., Reviewer SUs4, unfortunately did not, but left a brief comment to the global response that his or her major concern with the use of SAM mask refiner has been addressed (yet the reviwer seems not update the score). Overall, the AC cannot find any clear reason for rejecting the paper. Below I summarize the major concerns of the reviewers and how they are resolved.

- **Limited technical contribution (9zHz, SUs4)**: The reviewers considered the proposed method as a simple integration of image segmentation into MLLMs, which is incremental and engineering-oriented. The AC found that the authors' response to this commend is reasonable, and that the direction this paper proposes is novel enough, technical contributions may not be that significant though. Reviewer 9zHz still believes its technical contribution is limited, but voted for acceptance.
- **The use of SAM mask refiner (9zHz, SUs4, stRB)**: Reviewer 9zHz and Reviewer SUs4 raised concerns with potential fairness issue caused by the use of SAM mask refiner in comparisons with prior work, and Reviewer stRB considered that using the SAM mask refiner contradicts one of the main advantages the authors claimed in the paper (i.e., decoder-free architecture for semgnetation). The authors resolved this issue in the rebuttal by comprehensive experiments and analysis demonstrating that the proposed method works well with conventional post-processing techniques like dense CRF and achieves comparable performance even with no post-processing.
- **Lack of analysis on inference speed (9zHz)**: The analysis has been conducted and the results were reported in the revision.
- **Lack of comparisons with relevant methods (9zHz, SUs4)**: Reviewer 9zHz asked comparisons with specialist segmentation models, and Reviewer SUs4 requested comparisons with other MLLMs. The authors provided additional results following the suggestions and demonstrated that the proposed method is still superior in the additional experiments too.
- **Minor presentation issues (stRB)**:  Successfully addressed by the revision.

---

### Decision · Program_Chairs · 2025-01-22

Accept (Poster)